

**Reconciling different approaches to quantifying land surface temperature**
**impacts of afforestation using satellite observations**
Huanhuan Wang[1], Chao Yue[2,3*], Sebastiaan Luyssaert[4]
[1] College of Natural Resources and Environment, Northwest A&F University, Yangling,
Shaanxi 712100, P. R. China
[2] State Key Laboratory of Soil Erosion and Dryland Farming on the Loess Plateau, Northwest
A&F University, Yangling, Shaanxi 712100, P. R. China
[3] College of Forestry, Northwest A&F University, Yangling, Shaanxi 712100, P. R. China
[4] Department of Ecological Sciences, Faculty of Sciences, Vrije Universiteit Amsterdam,
Amsterdam 1081 HV, The Netherlands
Correspondence: Chao Yue, State Key Laboratory of Soil Erosion and Dryland Farming on
the Loess Plateau, Northwest A&F University, Yangling, Shaanxi 712100, P. R. China
E-mail: chaoyue@ms.iswc.ac.cn

## 17 Abstract

Satellite observations have been widely used to examine afforestation effects on local surface
temperature at large spatial scales. Different approaches, which lead potentially to differed
definitions of the afforestation effect, have been used in previous studies. The results were used
in climate model validation and were cited in climate synthesis reports, but large differences
existed in these results. Such differences were simply treated as observational uncertainty,
which can be an order of magnitude bigger than the signal itself. However, it remains unclear
whether these differences arise from methodological differences that can be reconciled or they
represent intrinsic uncertainty of land surface temperature change following afforestation. Here,





we provide a synthesis of three influential approaches (one estimates the actual effect and the
other two the potential effect) used in the literature and use large-scale afforestation over China
as a test case to examine whether the differences in the effects stem from methodological
differences. We found that the actual effect ($\Delta T_a$) often relates to incomplete afforestation over
a medium resolution satellite pixel (1km) for which LST is observed and that it increases with
the fraction of the pixel actually afforested (89% variation in $\Delta T_a$ being explained). One
potential effect approach quantifies the impact of quasi-full afforestation ($\Delta T_{p1}$), whereas the
other quantifies the potential impact of full afforestation ($\Delta T_{p2}$) by assuming a shift from 100%
openland to 100% forest coverage. An initial paired-samples *t*-test shows that $\Delta T_a < \Delta T_{p1} <$
$\Delta T_{p2}$ for the cooling effect of afforestation ranging from 0.07K to 1.16K. But when all three
methods are normalized for full afforestation, the observed range in surface cooling becomes
much smaller (0.79K to 1.16K). While potential cooling effects could indeed be realized
through full afforestation, they might not always be feasible, given other environmental
constraints such as the high water consumption of forests and competition for land usage.
Although potential cooling effects have a value in academic studies where they can be used to
establish an envelope of effects, they are misleading in a policy-making context where the actual
cooling effect better represents policy ambitions.

**Keywords**: surface temperature change, afforestation, actual effect, potential effect,
reconciliation, surface energy balance, China

1 Introduction

Afforestation has been and is still proposed as an effective strategy to mitigate climate change
because forest ecosystems are able to sequester large amounts of carbon in their biomass and



soil, slowing the increase of atmospheric $CO_2$ concentration (Fang et al., 2014; Pan et al., 2011).
Additionally, forests regulate the exchange of energy and water between the land surface and
the lower atmosphere through various biophysical effects, including radiative processes such
as surface reflectance, and non-radiative processes such as evapotranspiration and sensible heat
flux (Bonan, 2008; Juang et al., 2007). As the net result of the surface energy balance, land
surface temperature (LST) is widely used to measure the local climatic impact of afforestation
(Li et al., 2015; Winckler et al., 2019a).

Climate model simulations and site-level observations have been utilized to explore the impact
of forest dynamics on land surface temperature (Lee et al., 2011; Pitman et al., 2009; Swann et
al., 2012). However, afforestation impacts on local LST derived from models tend to be highly
uncertain as they are limited by the coarse spatial resolution of models and uncertainties in
model parameters and processes (Oleson et al., 2013; Pitman et al., 2011), while insights from
site-level assessments cannot be extrapolated to large spatial domains (Lee et al., 2011).
Alternatively, remote sensing-based LST products enable the assessment of local LST changes
due to forest dynamics on large spatial scales (Li et al., 2015; Shen et al., 2020).

A number of studies investigated the surface temperature impact of afforestation based on
satellite observations and they have been cited in high-level climate science synthesis reports
(e.g., *IPCC Special Report on Climate and Land* authored by Jia et al., 2019), although there
are large differences in afforestation impacts on LST among different methods. For example,
Alkama and Cescatti (2016), found a cooling effect of about 0.02K from afforestation in
temperate regions, while Li et al. (2015) reported a 0.27±0.03K 'potential' cooling from
afforestation in the northern temperate zone (20–50° N) based on the 'space-for-time' method.
In contrast, Duveiller et al. (2018) found a much stronger 'potential' cooling effect of 2.75K



for afforestation in the northern temperate region. While such differences were acknowledged
in a recent modelling study (Winckler et al., 2019b), they were simply treated as observational
uncertainty for climate model evaluation, with the uncertainty range being as big as, or even an
order of magnitude larger than, the afforestation effect. However, it remains unclear whether
these differences arise from methodological differences that can be reconciled or they indeed
represent the intrinsic uncertainty of the afforestation impact on LST.

Until now, studies using satellite data to investigate afforestation impact on surface temperature
mainly focused on three methods. The first method, termed the 'space-and-time' approach (Fig.
1, red box), aims to examine the actual, realized effect of afforestation ('actual effect') by
isolating the forest cover change effect from the gross temperature change over time in places
where forest cover change actually occurred (Alkama and Cescatti, 2016; Li et al., 2016a). The
second method, termed the 'space-for-time' approach (Fig. 1, orange box), compares the
surface temperature of forest with adjacent 'openland' (i.e., cropland or grassland) under similar
environmental conditions (e.g., background climate and topography) and estimates the
'potential effect' of afforestation if afforestation were to occur (Ge et al., 2019; Li et al., 2015;
Peng et al., 2014). Note that such effects are often quantified using medium-resolution land-
cover datasets (typical resolution = 1km), which do not necessarily represent 100% ground
coverage, and we therefore term such a potential effect a 'mixed potential effect'.

The third method, recently used by Duveiller et al. (2018), uses the 'singular value
decomposition' technique (Fig. 1 green box), which is claimed to extract the hypothetical LST
for different land-cover types by assuming a 100% coverage of the target cover type. The
afforestation effect on LST is then quantified as the difference between the LST of a pixel with
a hypothetical 100% forest coverage and the LST of an adjacent pixel with 100% openland



coverage. As with the second method, such an approach quantifies the 'potential effect' of
afforestation, but in this case, it quantifies the 'full potential effect' by assuming transitions
between land-cover types with 100% complete ground coverage. Given the aforementioned
methodological differences and, in particular, the different definitions of afforestation impact
on LST, confusion, if not misinterpretation, is expected when LST changes quantified using
these different approaches are used for model evaluation or policy recommendation.

This study develops detailed conceptual and methodological descriptions for each of the three
approaches, and uses large-scale afforestation over China as a case study to compare the three
approaches. We tested the following hypotheses: (1) The actual effect on LST increases with
the area that has actually been afforested, defined as afforestation intensity (or $F_{aff}$). (2) The
actual effect is smaller than the potential effects. (3) When extending $F_{aff}$ to a hypothetical value
of 100%, the actual effect approaches the potential effect. If proven, this third hypothesis
implies that the LST impacts from different approaches could be reconciled given the same
boundary condition of full afforestation. In that case, we then have a fourth hypothesis (4)
stating that changes in underlying biophysical processes including radiation, sensible and latent
heat fluxes that drive LST changes should also be reconciled among different methods. To keep
the focus on reconciling methodological differences, only changes in the daytime surface
temperature were considered in this study. Nevertheless, similar conclusions regarding the
different approaches are expected for nighttime surface temperature.





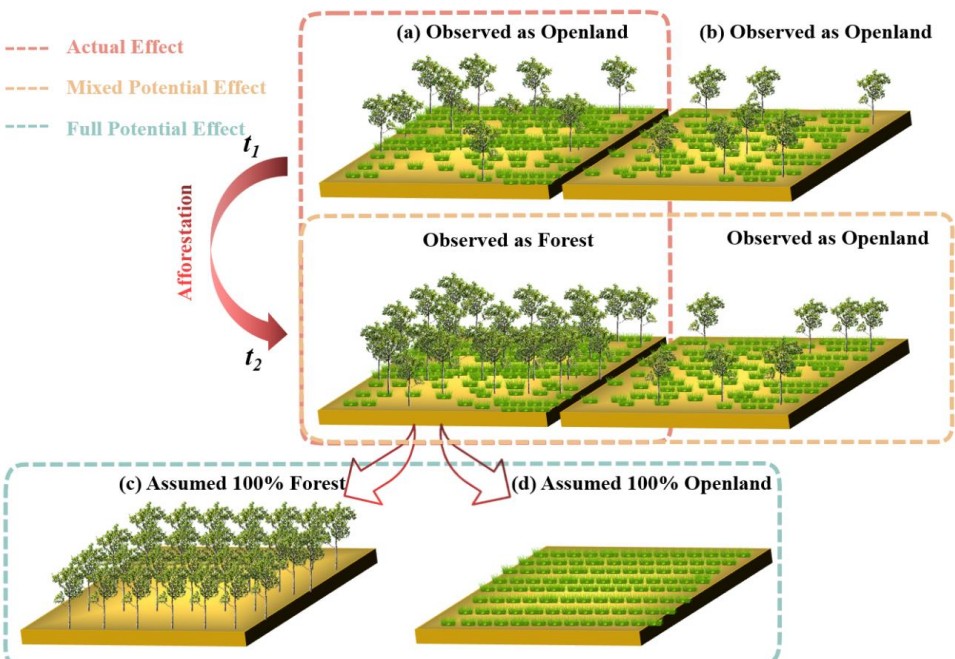

**Figure 1.** Illustration of the three approaches to quantifying the local surface temperature effect of afforestation. (a) and (b) represent two nearby pixels, both classified as openland at time $t_1$ by medium-resolution satellites (1km spatial resolution), with one of them classified as forest at time $t_2$ (i.e., having experienced afforestation) and the other unchanged. Note, neither of these pixels will have 100% complete coverage of either openland (i.e., grassland or cropland) or forest, but they will have been classified as either openland or forest by medium-resolution satellite products. (c) and (d) represent pixels with 100% forest or 100% openland coverage whose temperature can be derived from pixels of mixed land cover types by using the singular value decomposition (SVD) technique (Duveiller et al., 2018). The red dotted box describes the quantification of the 'actual effect' of afforestation ($\Delta T_a$) occurring from $t_1$ to $t_2$ by the 'space-and-time' method. The orange box represents the 'mixed potential effect' determined by hypothesizing potential shifts between openland and forest based on the 'space-for-time' approach ($\Delta T_{p1}$). The green box represents the 'full potential effect' of afforestation ($\Delta T_{p2}$)




derived by hypothesizing a transition from 100% complete openland coverage to 100%
complete forest coverage.

## 2 Methods

2.1 Three Approaches to Quantifying the Impacts of Afforestation on LST

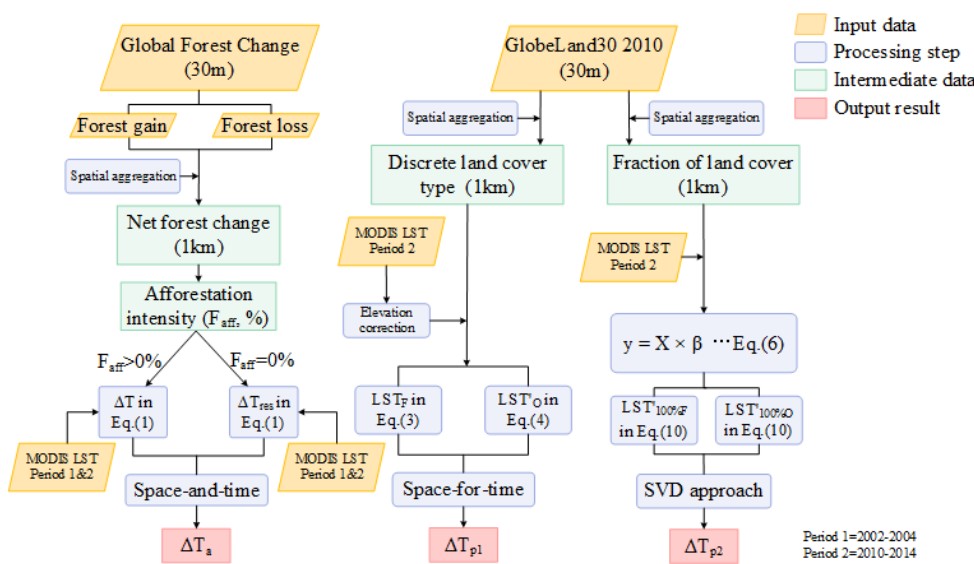


**Figure 2.** Schematic overview of the processing steps. The different output results correspond
to actual effect ($\Delta T_a$), mixed potential effect ($\Delta T_{p1}$) and full potential effect of afforestation
($\Delta T_{p2}$).

2.1.1 Actual Effect of Afforestation ($\Delta T_a$)

The 'space-and-time' approach assumes that the gross change in land surface temperature ($\Delta T$)
over a given time period during which afforestation occurred, contains both signals of
temperature change due to afforestation ($\Delta T_a$) and background temperature variation ($\Delta T_{res}$)
due to changes in large-scale circulation patterns (Alkama and Cescatti, 2016; Li et al., 2016a):



$$\Delta T = \Delta T_a + \Delta T_{res} \tag{1}$$
where $\Delta T$ is the gross temperature change during the period from $t_1$ to $t_2$ for the pixel under
study. $\Delta T$ can be calculated as the difference between $LST_{t2}$ and $LST_{t1}$, with $LST_{t2}$ being the
surface temperature after afforestation and $LST_{t1}$ being that before afforestation. It thus follows
that
$$\Delta T_a = \Delta T - \Delta T_{res} \tag{2}$$
$\Delta T_{res}$ can be approximated by averaging changes in surface temperature for those pixels
adjacent to the target afforestation pixel for which the forest cover remained constant between
$t_1$ and $t_2$ (i.e., $F_{aff} = 0\%$; section 2.2.2). Here, a search window of 11 km×11 km centered on the
afforestation target pixel was used to derive $\Delta T_{res}$. Afforestation pixels and adjacent control
pixels were both determined based on the net forest change between $t_1$ and $t_2$ using Global
Forest Change data (Fig. 2; section 2.2.2).

2.1.2 Mixed Potential Effect ($\Delta T_{p1}$)

The 'space-for-time' method relies on the 'space-substitute-for-time' assumption to obtain the
potential impact of afforestation on local temperature (Zhao and Jackson, 2014). By assuming
that forest and openland share the same environmental conditions (background climate,
topography, etc.) within a small spatial domain, the potential temperature effect of afforestation
is examined using the search window method with a window size of up to 40km×40km (Ge et
al., 2019; Li et al., 2015; Peng et al., 2014). Here, to be consistent with our 'actual effect'
approach, a more conservative window size of 11km×11km was used, smaller than that used in
the majority of previous studies (Ge et al., 2019; Li et al., 2015; Peng et al., 2014). In most
previous studies, existing medium resolution (1km) land-cover maps were used directly. Such
land-cover products rely on certain thresholds to classify satellite pixels into discrete land-cover



types. Given the widespread spatial heterogeneity in land-cover distribution, it is to be expected
that only in rare cases will these medium-resolution pixels have 100% coverage of a given land-
cover type. Therefore, when determined in this way, the potential effect of afforestation has
been named the 'mixed potential effect', in contrast to the 'full potential effect' which assumes
a potential transition between land-cover types of 100% coverage that we will focus on in the
next section.

To ensure consistency with the land-cover data used in the 'full potential effect' approach (i.e.,
the SVD method), the GlobeLand30 land-cover map was aggregated from its original resolution
(30m) to 1km resolution. The land-cover type assigned to a given 1km pixel during aggregation
was based on the land-cover type of the majority of the 30m sub-pixels within the 1km pixel,
to be consistent with the ideas behind the generation of medium-resolution land-cover products
(section 2.2.2). A 1km forest pixel was then chosen as the target pixel and put at the center of a
search window with dimensions 11km×11km. The 'mixed potential effect' of afforestation
($\Delta T_{p1}$) was defined as the difference between the temperature of the target pixel ($LST_F$) and the
average temperature of all the surrounding openland pixels within the window ($\overline{LST_O'}$):

$\Delta T_{p1} = LST_F - \overline{LST_O'}$                         (3)

where $LST_F$ is the surface temperature of the target forest pixel at $t_2$, and $LST_O'$ represents the
elevation-corrected surface temperature of openland pixels at $t_2$ within the search window.
Given our search window size, $\Delta T_{p1}$ could be biased by the elevation difference between the
target forest pixel and surrounding openland pixels. Therefore, a linear relationship was first
fitted between the observed openland temperature, $LST_O$, and the elevation of the openland
pixel ($Ele_O$). This fitted temperature lapse rate was then used to derive elevation-corrected
openland temperature $LST_O'$:





$$LST_O^{'} = LST_O + k \times \Delta Ele_{F-O} \tag{4}$$

where $\Delta Ele_{F-O}$ is the elevation difference between forest and openland pixels. The elevation is
available from NASA's Shuttle Radar Topography Mission (SRTM) data
(https://lpdaac.usgs.gov/products/srtmgl1v003/).

2.1.3 Full Potential Effect ($\Delta T_{p2}$)

The full potential effect represents the temperature change due to hypothesizing a shift from
100% openland to 100% forest coverage, and was determined here by employing the singular
value decomposition (SVD) method used in Duveiller et al. (2018). The SVD technique
assumes that the temperature observed for a pixel at 1km scale is a linear composition of the
temperatures of different land-cover types at a finer resolution (in our study at a 30m resolution).
For each 1km pixel, the observed temperature at 1km resolution can be written as the
composition of the temperature of each component land-cover type and its corresponding
fraction, based on the land-cover fractions derived from the 30m-resolution GlobeLand30 map
(section 2.2). The temperature of each type of land cover was assumed constant within a search
window of 11km × 11km. For each given search window, the following equations can be
obtained:
$$\begin{pmatrix} y_1 \\ \vdots \\ y_n \end{pmatrix} = \begin{pmatrix} x_{11} & \cdots & x_{1m} \\ \vdots & \ddots & \vdots \\ x_{n1} & \cdots & x_{nm} \end{pmatrix} \times \begin{pmatrix} \beta_1 \\ \vdots \\ \beta_m \end{pmatrix} \tag{5}$$

where n is the total number of 1km pixels within the window, after accounting for the elevation
difference (thus the maximum value of n is 121 given our 11km × 11km search window), m is
the number of land-cover types, $x_{ij}$ refers to the fraction of land-cover type $j$ in pixel $i$, $\beta_i$
refers to the temperature of land cover type $i$. To minimize elevation impacts, the linear





regression relationship for a given 1km pixel was included only when the elevation difference
between this pixel and the central pixel of the search window was smaller than 100m. Using
matrix notation, Eq. (5) can be simplified to:
$$y = X \times \beta \qquad (6)$$
where the matrix X contains land-cover fraction for the *n* 1km pixels as an explanatory variable,
the response variable y contains *n* LST observations, and the coefficient vector, β, contains the
regression coefficients which show temperatures of different land-cover types. Note that this
linear equation system cannot be readily solved simply because the matrix X is 'closed', i.e.,
by definition, the elements in each row of the matrix X add to 1. After removing the mean of
each column (Zhang et al., 2007), the matrix X was transformed, by applying the SVD
technique, to another matrix, Z, of reduced dimension (more details in Duveiller et al., 2018).
After this transformation, we have the following:
$$y = Z \times \beta^{'} + \varepsilon \qquad (7)$$
and the β' coefficient can be obtained from equation (8):
$$\beta^{'} = \left( Z^{t} Z \right)^{-1} Z^{t} y \qquad (8)$$
However, the β' vector calculated from the transformed matrix Z cannot directly provide
surface temperatures for corresponding land-cover types. To obtain temperatures for each land-
cover type by assuming 100% ground coverage, an identity matrix Y with its dimension equal
to the number of land-cover types must be constructed to represent the hypothetical case in
which each 1km pixel was 100% covered by a single land-cover type. The same transformation
as applied to the matrix X was then applied to Y, to obtain a transformed matrix Z'. Finally, the
predicted temperature ( $LST^{'}_{100\%}$ ) for each land-cover type assuming a 100% coverage was
calculated as:
$$LST^{'}_{100\%} = Z^{'} \beta^{'} \qquad (9)$$





For the central pixel of the local search window, $\Delta T_{p2}$ was defined as the difference between
the predicted $LST_{100\%}'$ for forest ($LST_{100\%F}'$) and openland ($LST_{100\%O}'$).
$$\Delta T_{p2} = LST_{100\%F}' - LST_{100\%O}' \tag{10}$$
More details, including an illustration of the SVD method, can be found in Fig. 7 in Duveiller
et al. (2018).

## 2.2 Dataset and Processing
### 2.2.1 The Test Case: Large-scale Afforestation over China

China was selected as the test case for addressing the important methodological issues in
quantifying land surface impacts of afforestation because afforestation is a key national strategy
for sustainable development and climate mitigation (Bryan et al., 2018; Qi et al., 2013).
According to the 8[th] National Forest Inventory conducted in 2013, China's afforestation area
has reached $6.9 \times 10^3$ million ha, accounting for 33% of the total global afforestation area (Chen
et al., 2019). Afforestation in China during 2000–2012 occurred mainly in regions with more
than 400 mm of precipitation per year (Fig. 3a), which is considered a threshold below which
there is a high risk of afforestation failing due to water limitation (Mátyás et al., 2013). China
covers a wide range of latitude from 3.9° N to 53.6° N and its forest ecosystems cover an
elevation range of 100m to 4000m. This wide range of climate zones, from tropical/subtropical
to temperate and boreal, make it highly suitable for our methodological analysis because the
impact of afforestation on LST might differ with latitude and background climate (Lee et al.,
2011; Alkama and Cescatti, 2016). Further justification for using China as a test case are the
strongly diverging published LST impacts of afforestation there, ranging from an actual effect
of -0.0036K decade[-1] by Li et al. (2020) to a potential effect of -1.1K by Peng et al. (2014).





2.2.2 MODIS Dataset and Preparation

In this study, the actual effect was estimated by combining the actual satellite-derived
afforestation for 2000 to 2012 (see Section 2.2.2) with satellite-based estimates of biophysical
variables for the periods 2002–2004 ($t_1$) and 2010–2014 ($t_2$). MODIS remote sensing products
for land surface temperature (MOD11A2), albedo (MCD43B3) and evapotranspiration
(MOD16A2) were used to characterize the biophysical effects (Table 1). The datasets were
regridded to harmonize spatial (1km) and temporal (annual) resolutions (Table 1).

The MOD11A2 product provides 8-day land surface temperature for 10:30 AM and 22:30 PM
from the Terra satellite, but here we focused on daytime surface temperature. Only valid LST
observations from the original data were used to compute the average daily values for a given
year. Years for which more than 40% of daily data are missing were excluded from the analysis.
Annual data were then aggregated to obtain the average annual temperature for periods $t_1$ and
$t_2$.

The MCD43B3 product provides white-sky and black-sky shortwave albedo at 16-day temporal
resolution (Table1). The observed white-sky albedo was used as the daytime albedo (Peng et
al., 2014). For evapotranspiration (ET), we used the ET band in MOD16A2, which includes
water fluxes from soil evaporation, wet canopy evaporation and plant transpiration. To calculate
the mean annual albedo and evapotranspiration for 2002–2004 ($t_1$) and 2010–2014 ($t_2$) we used
the same approach as used for LST.

2.2.3 Land-Cover Datasets and Processing





Two land-cover datasets were used in this study: the 'actual effect' approach was based on the
Global Forest Change (GFC) dataset, while the 'mixed potential effect' and 'full potential effect'
used the GlobeLand30 land-cover data (Table 1).

The SVD technique, used in the 'full potential effect' approach, requires a land-cover map with
a higher spatial resolution than the 1km spatial resolution of the LST data. The GlobeLand30
product, which is based on Landsat images, provides land-cover information for China at a 30m
resolution for the years 2000 and 2010 (Chen et al., 2015). Cultivated land and grassland in
GlobeLand30 were classified as openland. Discrete land-cover type information at 30m
resolution in 2010 was aggregated to obtain the area fractions of the different land-cover types
at 1km resolution, which were then used to construct matrix X in Eq. (5) (Fig. 2). Furthermore,
land-cover type information at the 1km scale was extracted, based on the vegetation type with
area fraction >50% for every 1km×1km window. This data was then applied in the 'space-for-
time' method to identify forest and openland (Fig. 2).

GlobeLand30 data is not suitable for detecting forest change (Zeng et al., 2021). The Global
Forest Change (GFC) data, however, provides forest gain and forest loss at a spatial resolution
of 30m between 2000 and 2012 and has been used for mapping global forest change (Hansen
et al., 2013). Forest loss events were identified for each year between 2000 and 2012, but forest
gain was only identified for the whole period, simply because forest loss is an abrupt change
which can be effectively identified over short time periods, but forest gain is a gradual change
which can only be confidently identified over longer time spans. Here, forest losses and gains
from GFC were aggregated at a 1km resolution to obtain net forest change (defined as forest
gain minus forest loss) during this period (Fig. 2). A positive net change indicates afforestation
and the area percentage of afforestation for the 1km pixel area was defined as $F_{aff}$. The land-



cover type of pixels with $F_{aff}$ = 0% was considered to be stable. This net forest-change
information was then used in the calculation of the actual afforestation-induced temperature
effect ($\Delta T_a$)(Fig. 2).

2.3 Decomposition of Changes in Surface Temperature

Changes in surface temperature following forest-cover change are the net result of changes in
underlying fluxes that collectively determine the land surface energy balance:
$$\Delta SW_{in} - \Delta SW_{out} + \Delta LW_{in} - \Delta LW_{out} = \Delta H + \Delta LE + \Delta G \qquad (11)$$
where $\Delta SW_{in}$, $\Delta SW_{out}$, $\Delta LW_{in}$, $\Delta LW_{out}$ are the changes in incoming and outgoing shortwave
and longwave radiation, respectively, and $\Delta H$, $\Delta LE$, and $\Delta G$ are changes in sensible heat flux,
latent heat flux and ground heat flux, respectively. All the terms of Eq. (11) are expressed in
$Wm^{-2}$.

Firstly, it can be reasonably assumed that $\Delta SW_{in} \approx 0$ and $\Delta LW_{in} \approx 0$, given that all three
approaches consider only local effects on surface temperature by following a comparison of
target pixels with surrounding control pixels, thus excluding feedbacks from, e.g., cloud
formation (Duveiller et al., 2018). Changes in reflected shortwave radiation can be derived as:
$$\Delta SW_{out} = SW_{in} \times \Delta \alpha \qquad (12)$$
where $SW_{in}$ is available from the CERES EBAF-Surface Product Ed 4.1 (Kato et al., 2018; Liu
et al., 2018) (Table 1), and $\Delta \alpha$ is the surface albedo change. To approximate $\Delta LW_{out}$, we used
its first order differential equation:
$$\Delta LW_{out} = \sigma(4\varepsilon_B T^3 \Delta T + \Delta \varepsilon_B T^4) \qquad (13)$$





where σ is Stefan-Boltzmann's constant ($5.67 \times 10^{-8}$ W m$^{-2}$ K$^{-4}$), T is daytime surface
temperature and ΔT is the afforestation impact on surface temperature. Surface broadband
emissivity, $\varepsilon_B$, is usually obtained from an empirical relationship (Zhang et al., 2019):
$$\varepsilon_B = 0.2122\varepsilon_{29} + 0.3859\varepsilon_{31} + 0.4029\varepsilon_{32} \qquad (14)$$
where $\varepsilon_{29}$, $\varepsilon_{31}$ and $\varepsilon_{32}$ are obtained from the estimated emissivity for bands 29 (8,400–8,700 nm),
31 (10,780–11,280 nm) and 32 (11,770–12,270 nm) in the MOD11C3 data (Duveiller et al.,

2018).


Latent heat flux change (ΔLE) refers to changes in the energy used for evapotranspiration (ET,
unit: mm d$^{-1}$), which can be obtained from the change in evapotranspiration (ΔET):
$$\Delta LE = \Delta ET \times 28.94 \text{ W m}^{-2}/(\text{mm d}^{-1}) \qquad (15)$$
Therefore, the sum of sensible heat change and ground heat change (ΔH+ΔG) can be calculated
as the difference between net radiation change and latent heat flux change (ΔLE) based on the
Eq. (11). The afforestation effects on albedo (Δα), $\varepsilon_B$ (Δ$\varepsilon_B$) and ET (ΔET) needed in the above
equations were calculated in a similar way to ΔT for each of the three different approaches as
described in section 2.1.

2.4 Statistical Analysis

Differences in the afforestation effects on LST of the three approaches were tested by
performing paired-samples $t$-tests between pairs of approaches. The paired-samples $t$-test was
used, rather than a normal $t$-test, to avoid the bias due to strong spatial heterogeneity in the LST
effects of afforestation that could occur if the values of all pixels had been pooled together for
a normal $t$-test. The pairing in the paired-samples $t$-test limits the analysis to only those pixels
shared by all three approaches. The test was made using the 'ttest_rel' method from the




'scipy.stats' package in Python. The Bonferroni correction was applied to adjust the
significance level (p-value) to mitigate the increasing the type I error when making multiple
paired-samples *t*-test, which in our case involves three pairs. The Bonferroni correction sets the
significance cut-off at α/k (with α as the p-value before correction and k as number of pairs). In
this study, with 3 hypotheses tests (i.e., 3 pairs) and an original significance level α = 0.05, the
adjust p-value is 0.0167. In order to investigate $\Delta T_a$ in relation to the afforestation intensity, a
linear regression was performed between $\Delta T_a$ and $F_{aff}$ using the ordinary least squares method.

Table 1 Summary of the datasets and their main characteristics

| Type | Dataset | Selected band | Resolution | Projection | Timespan |
|---|---|---|---|---|---|
| Forest change | Global Forest Change | Forest gain; Loss year | 30m, annual | WGS84 | 2000–2012 |
| Land-cover type | GlobeLand 30 | Land-cover type | 30m, — | UTM | 2000; 2010 |
| Land surface Temperature | MOD11A2 | Daytime temperature | 1km, 8days | sinusoidal | 2002–2004; 2010–2014 |
| Albedo | MCD43B3 | Albedo WSA shortwave | 1km, 16days | sinusoidal | 2002–2004; 2010–2014 |
| Incoming shortwave radiation | CERES | sfc_sw_down _all_mon | 1°, monthly | WGS84 | 2002–2004; 2010–2014 |



| Surface broadband emissivity | MOD11C3 | Emis_29; Emis_31; Emis_32 | 0.05°, monthly | sinusoidal | 2002–2004; 2010–2014 |
|---|---|---|---|---|---|
| Evapotranspiration | MOD16A2 | ET_500m | 500m, 8days | sinusoidal | 2002–2004; 2010–2014 |
| Elevation | SRTM30 | Be75 | 30m, — | WGS84 | — |


## 3 Results


### 3.1 Spatial Distribution of Afforestation and its Effect on Land Surface Temperature




Afforestation areas are mainly located in the northeast, southwest and south of China where
sufficient precipitation is available (Fig. 3a) and largely driven by afforestation of former
cropland or abandoned cropland, with a relatively small contribution from forest regeneration
or replanting following natural disturbance or timber harvest. One prominent feature of
afforestation in China is its small afforestation patch, with most afforested pixels (1km$^2$) having
an afforestation fraction of less than 30% (Fig. 3b). Pixels with an afforestation intensity below
10% account for 93% of the total number of pixels (Fig. 3b), representing 0.14 Mha or over
half (55.6%) of the total afforestation area (Fig. 3b).





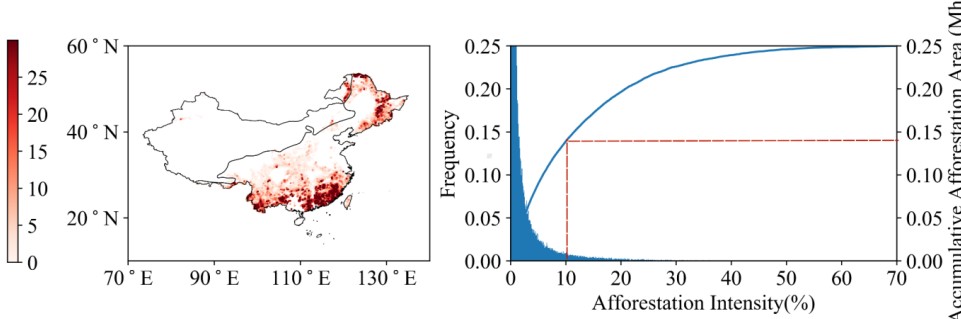


**Figure 3.** (a) Spatial distribution of afforestation intensity ($F_{aff}$) in China during 2000–2012.

The solid black line crossing China is the 400mm annual precipitation isoline. (b) Frequency

distribution of $F_{aff}$ and cumulative afforestation area with the increase in $F_{aff}$. The red dashed

line represents the cumulative afforestation area corresponding to $F_{aff}$ =10%.

Although all three approaches result in similar spatial and latitudinal patterns regarding

afforestation effects on LST (Fig. 4), their magnitudes differ substantially. The actual effect has

the lowest temperature change, followed by the mixed potential effect, with the full potential

effect showing the greatest temperature change (Fig. 4a–c). For the latitude range of 20–36° N

where afforestation effects show a dominant cooling effect, the full potential effect ($\Delta T_{p2}$)

reaches -1.75±0.01K, while the mixed potential effect ($\Delta T_{p1}$) was smaller at -0.96±0.00K, but

both of them were much larger than the actual effect ($\Delta T_a$) of -0.09±0.00K. Similarly, the full

potential effect ($\Delta T_{p2}$) showed the strongest warming effect (0.35±0.01K) in the area north of

48° N, stronger than the mixed potential effect (0.22±0.01K), and again the actual effect is the

smallest (0.07±0.01K). However, the three approaches largely converge regarding the latitude

where the effects change from a warming to cooling effect (Fig. 4d). Between 40° N and 48°

N, the afforestation effects are largely neutral, with the mean temperature change for the three

approaches being 0.07±0.01K ($\Delta T_a$=-0.01±0.01K; $\Delta T_{p1}$=0.11±0.01K; $\Delta T_{p2}$=0.12±0.01K).



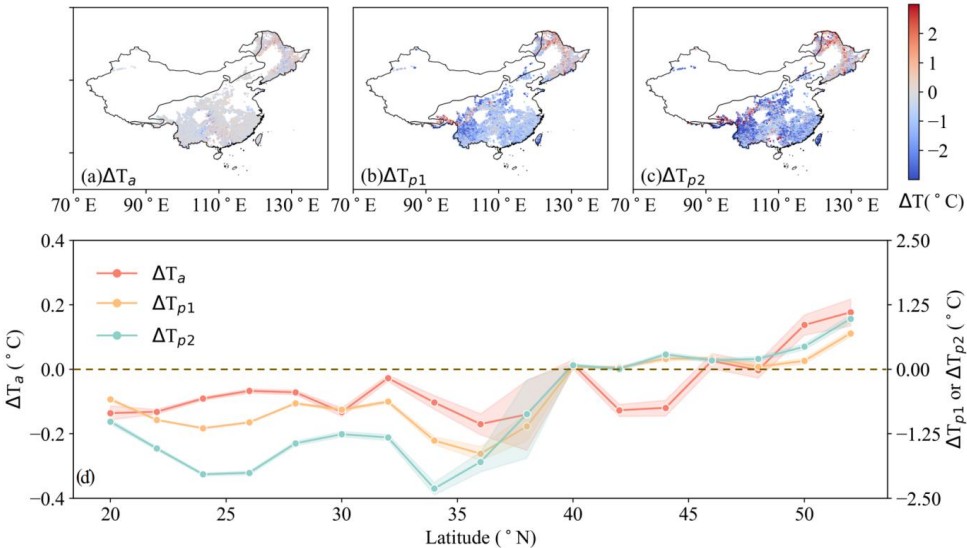

**Figure 4.** Afforestation effects on LST quantified by three approaches: (a) actual effect based on a 'space-and-time' approach ($\Delta T_a$), (b) mixed potential effect based on a 'space-for-time' approach ($\Delta T_{p1}$) and (c) full potential effect assuming a transition from 100% openland coverage to 100% forest coverage using the SVD method ($\Delta T_{p2}$). The solid black line crossing China is the 400mm precipitation isoline. (d) Zonal averages of the annual mean daytime LST change within 2° latitudinal bins, with shaded areas representing the standard errors (SE). Note that in panel (d), $\Delta T_a$ corresponds to the vertical axis on the left; $\Delta T_{p1}$ and $\Delta T_{p2}$ correspond to the vertical axis on the right.

3.2 Reconciling Temperature Effects of Afforestation

Even though the observed land surface temperature is assumed to be uniform for the 1km afforested satellite pixel, the underlying afforestation intensity varies substantially (Fig. 3a). This leads to our first hypothesis that for a 1km pixel, $\Delta T_a$ should be influenced by the area fraction that has been afforested within the pixel (i.e., afforestation intensity or $F_{aff}$). Indeed, the


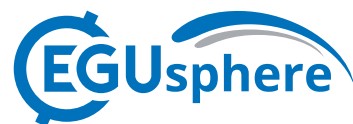

actual daytime surface cooling increases significantly with afforestation intensity (Fig. 5), with
a 0.079±0.017K (mean ± std) increase for each ten percent increase in $F_{aff}$.

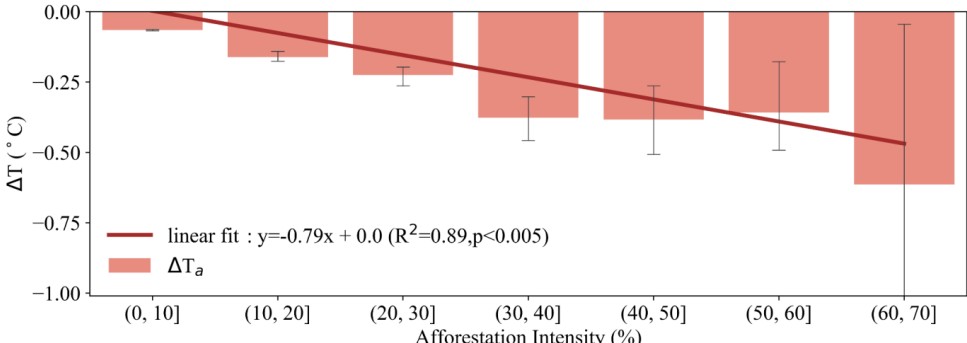


**Figure 5.** Changes in $\Delta T_a$ as a function of afforestation intensity ($F_{aff}$), defined as the fraction

of afforested area to the total pixel area at a 1-km resolution. Error bars indicate the standard
error of $\Delta T_a$ within each ten percent bin of $F_{aff}$. The red line represents the fitted linear
regression line between $\Delta T_a$ and $F_{aff}$.

The afforestation effects obtained from the three approaches were compared for each $F_{aff}$
interval (Fig. 6). When afforestation intensity is less than 60%, significant differences exist in
the temperature change obtained by the three approaches, with $\Delta T_a < \Delta T_{p1} < \Delta T_{p2}$. This result
confirms our second hypothesis that the actual effect is expected to be smaller than potential
effects. However, for pixels with relatively low $F_{aff}$, the mixed potential effect is found to be
smaller than the full potential effect, which is reasonable, but to our knowledge, has not been
reported before. When the afforestation intensity is greater than 60%, the significant difference
in cooling effect between the different approaches disappears, likely because afforestation
intensity, and the associated forest coverage at 1km resolution, reach high values, i.e., allowing
the 'potential' effects to actually be realized given a high enough afforestation intensity.





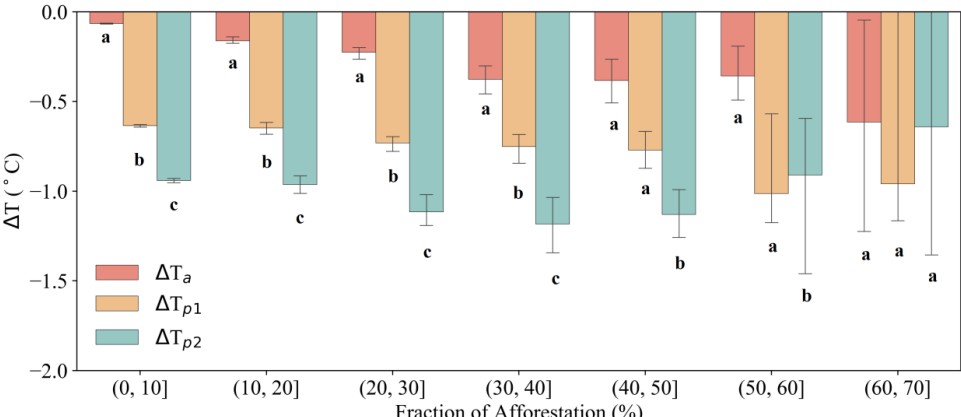

**Figure 6.** Comparison of ΔT for the three approaches for bins of afforestation intensity. Error bars are given as the standard error and different letters indicate that ΔT calculated by the two approaches concerned are significantly different with the adjust p-value after applying the Bonferroni correction with multiple paired-samples *t*-tests.

When considering the overall differences in ΔT from the three approaches, irrespective of the afforestation intensity, $\Delta T_a$ (-0.07±0.00K) over China was significantly lower than $\Delta T_{p1}$ (-0.63±0.00K), which is further significantly lower than $\Delta T_{p2}$ (-1.16±0.01K) ($p < 0.05$, paired-samples *t*-test, n= 96,058), once again confirming our second hypothesis (Fig. 7). Moreover, extrapolation of the relationship shown in Fig. 5 suggests that $\Delta T_a$ would reach -0.79±0.17K (mean ± std) if a 1km pixel was 100% afforested, which is conceptually comparable to the potential effects and it was indeed found to be higher than $\Delta T_{p1}$ but lower than $\Delta T_{p2}$. This result confirms our third hypothesis and demonstrates that the potential cooling effect could indeed be reached when a pixel is fully afforested.



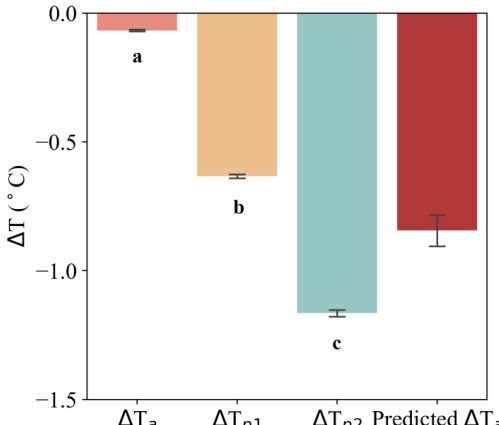

**Figure 7**. Comparison of $\Delta T$ for the three approaches, irrespective of the afforestation intensity. Error bars are given as the standard error and different letters indicate $\Delta T$ being significantly different (p = 0.0167, paired-samples *t*-test, n = 96,058). For comparison, the predicted $\Delta T_a$ with $F_{aff}$ reaching 100%, which is conceptually comparable with $\Delta T_{p1}$ and $\Delta T_{p2}$, is also shown.

## 3.3 Reconciling Changes in Surface Energy Fluxes by Afforestation

In order to investigate whether the underlying surface energy fluxes could be reconciled following the reconciliation of the LST changes, changes in surface energy fluxes due to afforestation were quantified using each of the three approaches, under the same boundary conditions as for full afforestation (i.e., changes following the 'actual effect' approach were extended for $F_{aff}$ = 100%). As illustrated in Fig. 8, changes in all the relevant surface energy fluxes under the three different approaches have the same direction, with similar magnitudes, confirming the reconciliation of the different approaches in terms of surface energy fluxes. More specifically, the three approaches converge on a reduction in reflected shortwave radiation ($\Delta SW_{out}$) of 0.56~1.23 W m$^{-2}$ due to the lower albedo of forest compared to openland (Figure A2). Meanwhile, emitted longwave radiation ($\Delta LW_{out}$) was reduced by 1.03~3.10 W




m$^{-2}$ and sensible and ground heat fluxes ($\Delta$H+$\Delta$G) reduced by 4.84~6.14 W m$^{-2}$. All these
reduced fluxes were offset by an increased latent heat flux of 7.99~8.41 W m$^{-2}$ ($\Delta$LE), the single
energy flux leading to surface cooling.

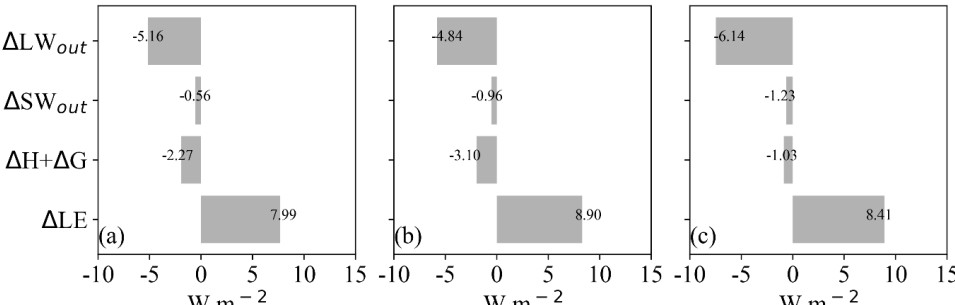


**Figure 8.** Afforestation-induced changes in surface energy fluxes (Wm$^{-2}$) following the three
approaches: (a) actual effect based on a 'space-and-time' approach, (b) mixed potential effect
using medium-resolution land cover maps based on a 'space-for-time' approach and (c) full
potential effect assuming a transition from 100% openland coverage to 100% forest coverage
using the SVD method. For each approach, changes were calculated for the reflected shortwave
radiation (SW$_{out}$), outgoing longwave radiation (LW$_{out}$), latent heat flux (LE) and the
combination of sensible and ground heat fluxes (H+G). No changes were assumed for incoming
shortwave and longwave radiation. Changes in energy fluxes for the 'actual effect' approach
have been adjusted to the condition of full afforestation (i.e., F$_{aff}$ = 100%) in a similar way as
for the 'predicted $\Delta$T$_a$' in Fig. 7, by fitting linear regressions between energy flux variables and
F$_{aff}$ (Figure A1).

4 Discussion

The three approaches (Li et al., 2015; Alkama and Cescatti, 2016; Duveiller et al., 2018) used
to quantify local surface temperature change following forest-cover change and presented with





details in this study, have been cited over 919 times in research papers (Web of Science,
December 2021) and in high-level climate science synthesis reports. Despite the apparently
large differences in temperature effect among them, to our knowledge, no studies have
examined whether these differences can be reconciled or whether they represent intrinsic
differences. This study fills that gap by comparing the three approaches for a single study case,
i.e., large-scale afforestation in China. China is highly suitable for the purpose of this study as
the size of an afforestation patch is, in general, smaller than the spatial resolution (1km) at
which the temperature effects of afforestation were conducted in the previous studies describing
the three approaches (Li et al., 2015; Alkama and Cescatti, 2016; Duveiller et al., 2018). Hence,
the difference between the actual and potential temperature effects is expected to be large.

Indeed, we found surface cooling following afforestation was much less when estimated as the
actual effect ($\Delta T_a$) compared to the potential effects ($\Delta T_{p1}$ and $\Delta T_{p2}$). This lower $\Delta T_a$ has been
attributed to incomplete afforestation at a 1km resolution, at which potential effects are
quantified by assuming complete afforestation (i.e., a complete shift from openland to forest).
Consistent with our first hypothesis, the afforestation fraction at a 1km resolution explained 89%
of the variation in $\Delta T_a$, making it a key determinant of the surface cooling following
afforestation (Fig. 5). This finding is in line with the fundamental fact that surface temperature
can be largely treated as an extensive variable: a variable whose whole pixel value of a given
property is strongly determined by the area fractions of its different components, with each
component having a unique value for the given property. The observation that surface
temperature is an extensive variable served as the theoretical foundation for the SVD technique
to derive the full potential effect (Duveiller et al., 2018).



Modelling (Li et al., 2016b) as well as satellite-based (Alkama and Cescatti, 2016) studies have
found that temperature change after afforestation (or deforestation) is highly sensitive to the
fraction of the model grid cell or satellite pixel that is subjected to afforestation (or
deforestation), echoing our finding that $\Delta T_a$ significantly changes with $F_{aff}$. In addition, we
provide strong evidence in support of our third hypothesis that when $F_{aff}$ reaches 100%, the
expected actual effect is comparable to the potential effects (Fig. 7). This finding shows that
the three approaches compared in this study are consistent when the same boundary condition,
i.e., full afforestation, is applied, and demonstrates that all three methods are mutually
compatible. It is, therefore, the basis of the reconciliation of the three approaches. Meanwhile,
it highlights that the actual afforestation area must be considered when evaluating climate
mitigation effects of afforestation.

Our results also show that the mixed potential effect ($\Delta T_{p1}$) is smaller than the full potential
effect ($\Delta T_{p2}$) (Fig. 6, Fig. 7). We suspect that this phenomenon likely also relates to the
incomplete forest coverage for the identified forest pixels at the 1km scale used in the 'space-
for-time' analysis, because a threshold value of 50% forest cover was used when upscaling the
30m land-cover map to 1km resolution. This threshold, however, is consistent with the
commonly applied value in land-cover classification based on medium resolution satellite
images, such as MCD12Q1, which uses a tree coverage value of 60% to identify forest pixels
(Sulla-Menashe and Friedl, 2018). For the purpose of comparison, we also calculated the mixed
potential effect based on the MCD12Q1 land-cover map but using the same LST data. The
result shows that potential effects derived using MCD12Q1 data versus those derived using
spatially upscaled GlobeLand30 data are almost identical (Figure A3), lending credibility to our
estimated $\Delta T_{p1}$ in comparison to previous studies using MODIS land-cover data (Li et al., 2015).
Progressively increasing the forest-cover threshold from 50% to 90% steadily increases $\Delta T_{p1}$





from  -0.62±0.02K  to  -0.75±0.02K  (Figure A4).  Further  increasing  the  thresholds  used  to
identify 1km-resolution openland pixels from 50% to 90% increases $\Delta T_{p1}$ from -0.63±0.00K to
-1.10±0.02K (Figure A5), bringing $\Delta T_{p1}$ even closer to $\Delta T_{p2}$ (-1.16±0.01K). This adds further
support to the compatibility of the three approaches given the same boundary condition, i.e.,
the complete transformation from full openland to full forest coverage.

Previous analyses have documented latitudinal patterns of surface temperature change induced
by afforestation (Alkama and Cescatti, 2016; Li et al., 2015, 2016a; Peng et al., 2014). When
comparing the three approaches for a single case study, consistent latitudinal patterns of local
surface temperature effects following afforestation are observed (Fig. 4). Notably, all three
approaches show a warming effect in the northern high latitudes and an opposite cooling effect
in the southern low latitudes, with a largely neutral effect in the 40–48° N latitude band,
providing further evidence that the three approaches are compatible. In particular, although the
three  approaches  used  different  land-cover  maps,  they  derived  consistent  LST  impacts
following afforestation, which highlights that the reconciling provided in this study is rather
robust and is unlikely dependent on the land cover datasets used.

In addition to the reconciliation of the land surface temperature change, we checked and
confirmed that the changes in surface energy fluxes that underlie and drive the changes in
surface temperature are compatible under the boundary condition of full afforestation. This
finding confirms the inherent consistency in the three approaches and clarifies the reasons
behind the apparent discrepancies in existing studies as discussed in the introduction.
Nonetheless, when it comes to the biophysical impacts of afforestation in the real world, our
findings have far-reaching implications. Although the 'potential effect' of afforestation could
indeed be reached, the condition of full afforestation might not be feasible in reality. For





example, a complete afforestation of semi-arid Loess Plateau in the northwest of China is
predicted to generate a surface cooling effect of 2.40±0.07K, but substantial afforestation efforts
over the past 4 decades in that region have only realized a cooling of 0.11±0.01K as measured
by the 'actual effect'. Because of greater water consumption by forest compared to openland
and the need to maintain land area for food production, achieving the full cooling potential may
not be feasible (Huang et al., 2018; Liu and She, 2012; Liang et al., 2019).

Whereas potential cooling effects have a value in academic studies where they can serve to
establish the envelope of effects, they are misleading in a policy-making context where the
actual cooling effect better represents policy-ambitions. The analog could also be made for the
effects of the surface energy impacts of afforestation. Taking 10% as the afforestation intensity
threshold to compare the cumulative surface energy effect between the actual and potential
approaches, actual cumulative biophysical changes (5.06 EJ) for 2000–2012 are much smaller
than mixed potential changes (20.13 EJ) and full potential change (19.02 EJ) (Figure A6). Again,
this shows that simply using the potential effects for policy making or evaluation risks greatly
overestimating the biophysical effects of afforestation.

## 5 Conclusions
In this study we provided a synthesis of the three influential methods used to quantify
afforestation impact on surface temperature change and provided evidence that these different
methods could in fact be reconciled. The actual effect of surface temperature change following
afforestation was highly dependent on the intensity of afforestation ($F_{aff}$), which explained 89%
of the variation in $\Delta T_a$. With the common boundary condition of full afforestation being applied,
differences in afforestation impacts on LST reported by the three methods in previous studies
greatly reduced, showing that simply treating these differences as uncertainty is incorrect and





could greatly overestimate the uncertainty. In other words, when full afforestation is assumed,
actual effect approaches the potential effect, demonstrating the effectiveness of the 'space-for-
time' approach and that potential cooling effect of afforestation could be indeed realized.
However, due to the environmental constraints such as water availability and land scarcity,
large-scale full afforestation might not always be feasible. In this case, potential effect would
provide an envelop of the effects of afforestation but only the actual effect has a direct policy
relevance in evaluating the climate effects of afforestation projects.




















**Appendix A**

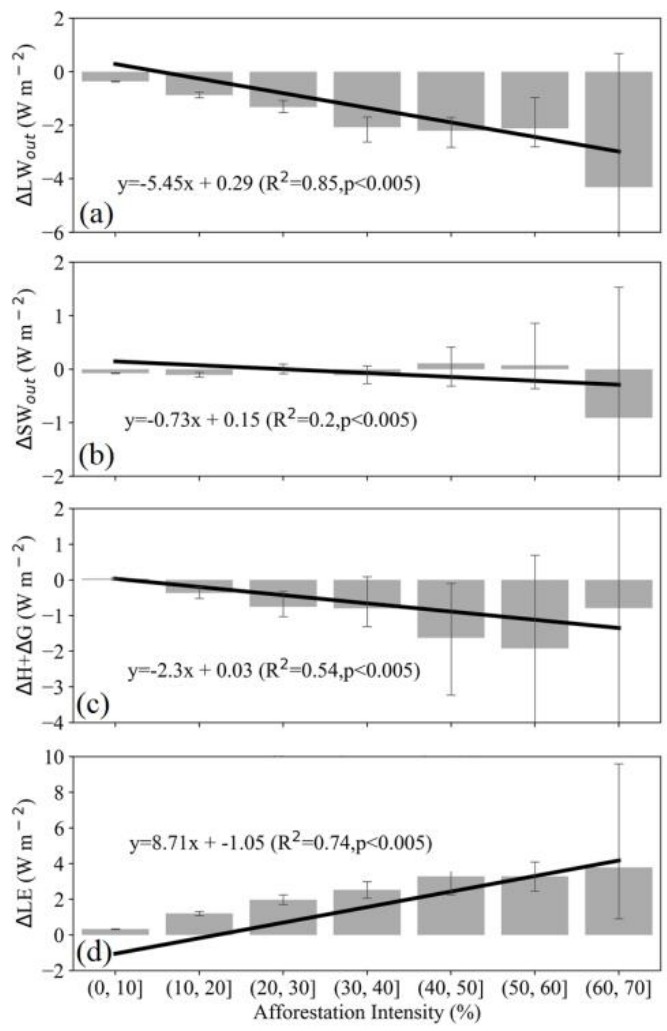


**Figure A1.** Changes of actual effect in (a) $\Delta LW$, (b) $\Delta SW$, (c) $\Delta H + \Delta G$ and (d) $\Delta LE$ (W m$^{-2}$)
as a function of afforestation intensity ($F_{aff}$) following the 'actual effect' approach. Error bars
indicate the standard error within each ten percent bin of $F_{aff}$. The solid black lines represent
the fitted linear regression line between each energy flux variable and $F_{aff}$.



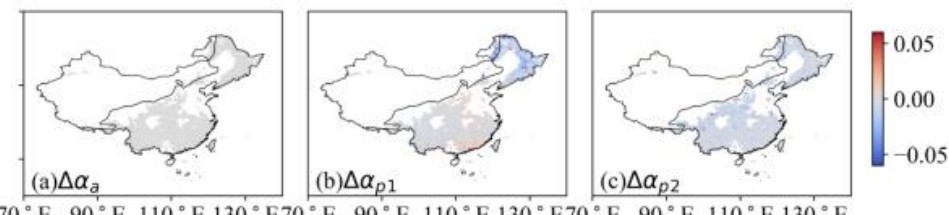

**Figure A2.** Spatial distribution of afforestation-induced changes in albedo (α) over China from three approaches: (a) Actual albedo change following afforestation based on 'space-and-time' method ($\Delta\alpha_a$), (b) mixed potential albedo change using medium-resolution land-cover maps based on 'space-for-time' approach ($\Delta\alpha_{p1}$) and (c) full potential effect ($\Delta\alpha_{p2}$) based on SVD approach.

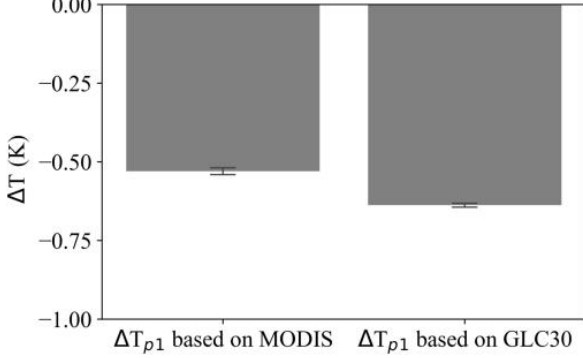

**Figure A3.** The mixed potential effects ($\Delta T_{p1}$) obtained based on MODIS land-cover data (MCD12Q1) and the land-cover distribution map defined at the threshold of 50% GlobeLand30 (GLC30) at 1 km resolution.




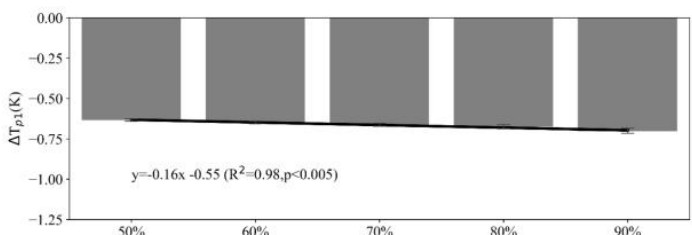


**Figure A4.** The influence of the forest-cover threshold applied to the land-cover map
underlying the estimation of the mixed potential effect ($\Delta T_{p1}$).

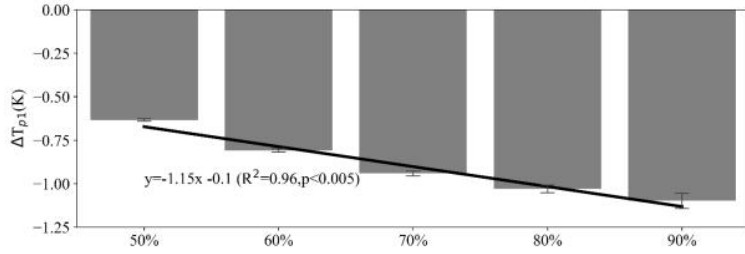


**Figure A5.** The influence of the openland-cover threshold used to identify a 1km pixel as
openland in the estimation of the mixed potential effect ($\Delta T_{p1}$).

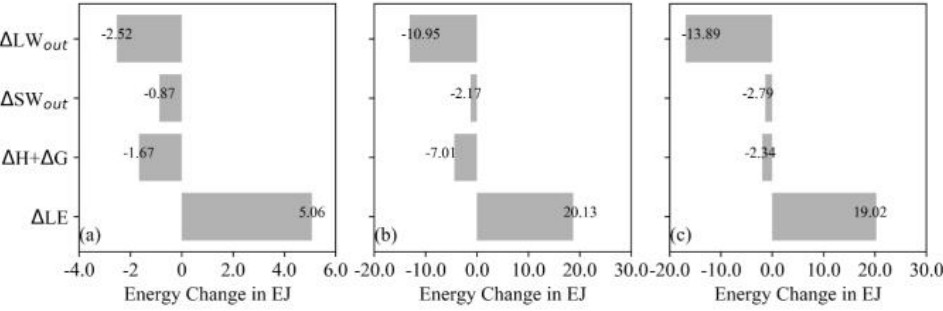


**Figure A6.** Afforestation-induced cumulative changes in surface energy fluxes (exaJoules) in
China for the period 2000–2012 following the approaches of (a) actual effect, (b) mixed
potential effect and (c) full potential effect.



**Data availability**

All datasets used in this study are summarized in Table 1 and are openly available. Albedo, transpiration and surface temperature can be accessed at (https://modis.gsfc.nasa.gov/data/). Global Forest Change is available from https://earthenginepartners.appspot.com/science-2013-global-forest/. The land-cover type dataset (GlobeLand30) can be downloaded from http://www.globallandcover.com/. Incoming shortwave radiation can be accessed at https://ceres.larc.nasa.gov/data/. The elevation is available from NASA' s Shuttle Radar Topography Mission (SRTM) data (https://lpdaac.usgs.gov/products/srtmgl1v003/). Intermediate data and scripts used to generate the results in this study are available from the corresponding author upon reasonable request.

**Author contributions**

Chao Yue and Sebastiaan Luyssaert designed the study. Huanhuan Wang conducted the analysis. All three authors contributed to writing and revision of the text.

**Competing interests**

The authors have the following competing interests: At least one of the (co-)authors is a member of the editorial board of Biogeosciences.

**Acknowledgments**

This study was supported by the Strategic Priority Research Program of the Chinese Academy of Sciences (grant no. XDB40020000) and by the National Natural Science Foundation of China (grant no. 41971132).

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
