# Peer review of "Reconciling different approaches to quantifying land surface temperature impacts of afforestation using satellite observations"

_EGUsphere, 2022_

## Author Comment (AC1)

Ref.: MS. bg-2022-317 Biogeosciences Reconciling different approaches to quantifying land surface temperature impacts of afforestation using satellite observations

General comments:

This study conducted an interesting research about three influential approaches in evaluating the climatic effects induced by afforestation over China. So far, no such studies have ever compared the three methods simultaneously and investigated the underlying mechanisms that lead to their discrepancies and more importantly, whether the discrepancies can be mitigated or reconciled. I'm happy to see that the authors filled this knowledge gap and gave us a good reference. As far as I know, in previous studies involving both the actual and potential effects (Li Yan, 2016, JGR-A, Shen Wenjuan, 2019, AFM), the two effects, characterized by LST changes (or cooling) were comparable and consistent in magnitude. As a result, their discrepancies attracted less attention. Fortunately, this research emphasized this point by applying the afforestation experiment over China. Coincidentally, I have a pending research (in prepare for subscription) in support of the result (actual effect is largely less than potential effect) in this study.

Overall, I appreciate the authors' efforts to put this question forward and gave a good demonstration.

We appreciate the comments which will help us to improve the manuscript. We are also glad that this researcher reaches a similar conclusion. Please find below the original comments (in black) and our responses (in blue).

Specific Comments:

(1) The distribution of sample grids about the actual and potential effect were not shown. Maybe you can display them in Supplemental Materials, like Peng Shushi et al., 2014, PNAS did.

We will add the distribution of sample grids of the actual and potential effects in the supplemental material in the revised manuscript (MS) (shown below in Fig. R1).

[Figure]

Figure R1. The distributions of the original sample pixels (1km resolution) for (a) actual effect and (b) potential effect.

(2) Line 313: Please explain why GlobeLand30 is not suitable for detecting forest change, instead of just citing Zeng et al., 2021.

To address this comment, we showed in Fig. R2 (below) that GlobeLand30 failed to capture the large-scale forest gain in Southeast China (range of 110°-120°E, 25°-30°N) from 2000 to 2010. Therefore, we did not directly use the GlobeLand30 dataset to detect forest change. Nonetheless, we decide not to include this Fig. R2 in revised MS in order to avoid the redundancy of the Method section.

[Figure]

Figure R2. The spatial distribution of forest gain pixels (1km resolution) detected based on GlobeLand30 between 2000-2010.

(3) When computing the mixed and full potential effects, what threshold did the authors use to define a 1-km pixel as an afforested pixel using the GFC data? In addition, the method to

process land cover data (GlobeLand30) seems to be ambiguous, since Line 189 described using the majority method to aggregate 30 m to 1km, but Line 309-310 mentioned "vegetation type with area fraction > 50% for every 1km×1km window". In my opinion, majority does not equal > 50%. For instance, one land cover type (i.e., cropland) accounts for 30% can also be designated as the dominated type as long as 30% is the largest area fraction.

For the first question, 1-km pixels with the net forest cover gain > 0 according to the GFC dataset were defined as afforested pixels. We will modify sentences in the Methods section in the revised MS to more clearly define afforestation pixels.

Regarding the second question, we acknowledged that the expression of "majority method" in the MS was misleading. 50% was set as a fraction threshold to define the land cover types based on GlobeLand30. We will revise the related description in MS.

(4) Line 311. What dataset did forest and openland stem from? Based on the early description, forest was only from GLC data and openland only from Globeland30. Please give a clear declaration here. Once more, it's important to clearly elucidate the criterion to define the afforested 1-km pixel when aggregating 30-m pixels. If the authors used 50% as the threshold, then the bars below 50% in Figure 6 seem to be unreasonable because pixels with afforestation fraction below 50% was not afforestation anymore. But if using a lower threshold, would the 1-km pixel stay as an afforestation pixel? Please, give an explicit and consistent explanation.

For the first question, forest and openland stem from the generated land cover map based on GlobeLand30. Forest pixels in this map (Fig. R1b) were selected as samples to obtain potential effects.

For the second question, pixels where forest gained (i.e., afforestation fraction >0%) as detected from GFC (Fig. R1a) were selected to derive the actual effect. Thus, the afforestation fractions in Fig. 6 are reasonable. These two points have been described in the Method section, so we prefer not to make any modifications to MS.

(5) When collecting the sample pixels, did the authors consider the impact of water pixels? As far as I know, the common method is to abandon the grids in which water pixels account for more than a fraction (5% or 10% or 15%...).

[Figure]

Figure R3. Histogram of water cover fraction of all samples in this study.

In the previous MS, the impact of water cover impact was not considered. As suggested, we took into the "water fraction" and checked the water cover fraction of research pixels based on the land cover fraction map (Fig. 2 in MS) from GlobeLand30. As shown in Fig. R3, the water fraction among all samples in this study is less than 10%, with almost 95% of samples containing no water area (water fraction=0%). In this way, we believe that removing samples where the water fraction is >5% or 10% may have less effect on the findings of this study, therefore we prefer not to change research samples in MS.

(6) Section 2.4, I wonder about the significance and necessity of using Bonferroni correction in this study. Many audiences including me seem not to be familiar with this operation. The authors may give a more detailed explanation.

We performed paired samples t-test to examine the differences in the afforestation effects on land surface temperature (LST) based on three approaches, which involved three hypotheses, i.e., $\Delta T_a = \Delta T_{p1}$; $\Delta T_a = \Delta T_{p2}$; $\Delta T_{p1} = \Delta T_{p2}$ in this study. Here, we employed three paired comparisons to test these hypotheses.

For each comparison, if we use significance level (p-value) =0.05 to determine that the means

of a pair of conditions (e.g., $\Delta T_a$ and $\Delta T_{p1}$) are statistically different from each other, we will have a 5% chance of committing a Type I error when we reject the null hypothesis ($H_0$: $\Delta T_a=\Delta T_{p1}$). When conducting three comparisons, the possibility of committing a Type I error for comparisons can be estimated as 0.05×3. Bonferroni correction was applied in this study to adjust the p-value to mitigate the increasing type I error when making multiple paired-samples t-tests (Lee and Lee, 2018; UC Berkely, 2008). We will cite related literature on Bonferroni correction in the Method section in the revised MS, but the explanation here will not be added to the revised MS.

(7) Figure 6. When the fraction of afforestation reached (50, 60], why the mixed potential effect exceeded the full potential effect. It seems strange and no explanation about this phenomenon was seen. In addition, the significant linear trend can be found for actual effect (as displayed in Figure 5), but it seems that this significant trend was not found in mixed potential especially the full potential effect. May the authors give an explanation about this?

As for the first question, we checked the data processing script and found Fig. 6 in the original manuscript was wrong and will be corrected as Fig. R4. In Fig. R4, when the fraction of afforestation reached (50, 60], the mixed potential effect is still smaller than the full potential effect. The relevant description in the Results section of our original manuscript still applies and will not be changed.

For the second question, $\Delta T_{p1}$ and $\Delta T_{p2}$ are grouped into intensity bins not because they have any relation to the afforestation intensity bins. Conceptually, they represent a complete shift from openland to forest, so it's expected they do not show any trend here.

[Figure]

Figure R4. Comparison of $\Delta T$ for the three approaches for bins of afforestation intensity.

(8) The reconciliation was reached when increasing the fraction to 100% for the actual effect. But why the fraction increase (through linear extrapolation) was only implemented for actual effect rather than both actual and mixed potential effect. It seems unfair because the author compared the 100% fraction-based actual effect with not 100% based (mixed) potential effect.

We believe this comment is related to Comment #7. From the concept of $\Delta T_{p1}$ and $\Delta T_{p2}$, it does not make sense to make regression between $\Delta T_{p1}$ and $\Delta T_{p2}$ and $F_{aff}$.

(9) What is the difference between Figure 8 and Figure A6? Mean values of all grids for Figure 8 and gross values of all grids for Figure A6? Do the cumulative biophysical changes only refer to delta_LE? Because the numbers in Line 586-587 corresponded to delta_LE in Figure. A6.

For Figure 8, afforestation-induced changes in surface energy fluxes referred to the flux change per unit area (W m$^2$), whilst cumulative surface energy effect ($f_{cum}$) in Figure A6 referred the sum of the flux change (J) from all the samples after considering the forest change area (m$^2$). More specifically, the cumulative surface energy change ($f_{cum}$) can be calculated from the equation R1:

$$f_{cum} = \sum_{i=1}^{i=n}(area_i \times F_i) \qquad (R1)$$

where $F_i$ is the flux change in per unit area (W m$^2$) for pixel $i$, n is the total number of samples, and area$_i$ is the forest change area in pixel $i$. This part will be added in Supplementary Material.

(10) Uncertainty about the Global Forest Cover dataset should be discussed. References can be found in recent papers published by Dr. Zeng Zhenzhong.

We assumed the reviewer was most likely referring to the GFC accuracy discussion. Comparing forest gained area from GFC to forest area statistics reported in Forest Resource Assessment (FRA), LiDAR detection (Geoscience Laser Altimeter System), and MODIS NDVI time series, the GFC product demonstrated a global accuracy of greater than 99% (Hansen et al., 2013). Chen et al. (2020) applied the global land cover validation data from the United States Geological Survey (USGS) to evaluate the accuracies of the selected land cover datasets while the correlations between the GFC dataset and the validation data were the highest (0.77). Zeng et al. (2021) also demonstrated that GFC can achieve an overall accuracy of 98.4% in Southeast

Asia. We will add some description of GFC uncertainty in the Methods section.

(11) The reasons leading to the discrepancies between actual and potential effects were not considered and discussed thoroughly.

1) Actual effect was calculated using the LST data from two years (target and reference year), but the potential effect used the LST from the same year (2012 in this study).

2) When computing the actual effect, the control pixels were constant or stable unchanged forests, however, as for potential effect, the reference pixels were cropland or grassland pixels.

3) Even though the author adopted the same sample pixels (same locations) for the three approaches, the inherent afforestation fraction was not consistent because different criteria were adopted.

Please give a detailed explanation and discussion about the above aspects.

We believe the point (1) and (2) refer to the method difference of "space-and-time" between "space-for-time". We will add a detailed description in the Supplementary Material to further explain the methodological differences to clarify the results of our research are not limited to fraction differences. While for point (3), we have never claimed that the inherent afforestation fraction for three methods is consistent, and the specific reasons can be found in our responses to comment #7. Based on previous research (Li et al., 2016), it could be reasonably suspected that the differences in estimated land surface cooling by afforestation by different approaches could be potentially due to the afforestation fraction, but this suspect was only proved until our current work.

**References used in the responses:**

Chen, H., Zeng, Z., Wu, J., Peng, L., Lakshmi, V., Yang, H., and Liu, J.: Large Uncertainty on Forest Area Change in the Early 21st Century among Widely Used Global Land Cover Datasets, Remote Sensing, 12, 3502, https://doi.org/10.3390/rs12213502, 2020.

Hansen, M. C., Potapov, P. V., Moore, R., Hancher, M., Turubanova, S. A., Tyukavina, A., Thau, D., Stehman, S. V., Goetz, S. J., and Loveland, T. R.: High-resolution global maps of 21stcentury forest cover change, science, 342, 850–853, 2013.

Lee, S. and Lee, D. K.: What is the proper way to apply the multiple comparison test?, Korean J Anesthesiol, 71, 353–360, https://doi.org/10.4097/kja.d.18.00242, 2018.

Zeng, Z., Wang, D., Yang, L., Wu, J., Ziegler, A. D., Liu, M., Ciais, P., Searchinger, T. D., Yang, Z.-L., Chen, D., Chen, A., Li, L. Z. X., Piao, S., Taylor, D., Cai, X., Pan, M., Peng, L., Lin, P., Gower, D., Feng, Y., Zheng, C., Guan, K., Lian, X., Wang, T., Wang, L., Jeong, S.-J., Wei, Z., Sheffield, J., Caylor, K., and Wood, E. F.: Deforestation-induced warming over tropical mountain regions regulated by elevation, Nature Geoscience, 14, 23–29, https://doi.org/10.1038/s41561-020-00666-0, 2021.

Duveiller, G., Hooker, J., and Cescatti, A.: The mark of vegetation change on Earth's surface energy balance, Nat Commun, 9, 679, https://doi.org/10.1038/s41467-017-02810-8, 2018.

Li, Y., Zhao, M., Mildrexler, D. J., Motesharrei, S., Mu, Q., Kalnay, E., Zhao, F., Li, S., and Wang, K.: Potential and Actual impacts of deforestation and afforestation on land surface temperature: IMPACTS OF FOREST CHANGE ON TEMPERATURE, J. Geophys. Res. Atmos., 121, 14,372-14,386, https://doi.org/10.1002/2016JD024969, 2016.

UC Berkely. Spring 2008 - Stat C141/ Bioeng C141 - Statistics for Bioinformatics

---

## Author Comment (AC2)

Ref.: MS. bg-2022-317 Biogeosciences Reconciling different approaches to quantifying land surface temperature impacts of afforestation using satellite observations

**Reviewer#1**

General consideration:

The manuscript "Reconciling different approaches to quantifying land surface temperature impacts of afforestation using satellite observations" by Wang et al presented thoughtful analyses regarding three different types of temperature effects of forestation that appeared in the literature (potential vs actual) and trying to explain the causes of the different magnitudes. The research is a nice addition to the literature on this topic as it is helpful to clarify the interpretation of different results.

We thank Reviewer#1 for the general positive comments and confirming the additional value of our work to the existing literature. Please find below our detailed responses to the review comments, with original comments in black and our responses in blue.

**Major comments:**

1. First, I disagree with the authors' interpretation of these results and the claim that the causes of the different estimates are unknown. On the contrary, spatial scale or fractions of forest change matters for interpreting the temperature impact, which has been considered in previous studies. Taking the influential work cited by the authors as an example:

In Alkama 2016, the fraction of forest cover change is explicitly taken into account, and the results clearly indicated that the temperature effect depended on the fraction of change.

In Li 2016, the fractional dependency has been reported: "It should be noted that the estimated impacts also depend on the thresholds used to define forest cover change, as discussed in section 2.2. The sensitivity analysis shows that a higher threshold to define forest change leads to stronger impacts on temperature."

In Duveiller 2018, they used the temperature effect of 100% conversion to avoid the influence of fractional changes.

The strength of this work is that it explicitly addressed this question. Perhaps the authors could consider an alternative title better reflecting this point.

Maybe some improper expressions (very likely they could be lines 25 and 81) in our original manuscript (MS) made the reviewer conclude that we have claimed that "the causes of the different estimates are unknown." But in fact, we did not explicitly claim this. Instead, we suspected that different forest cover changes in different approaches could potentially influence the LST change, which is one of the motivations for this study.

The reviewer seems to imply that the fraction of forest cover change was known as the cause of different magnitudes of LST change in previous studies. But this is not true. We acknowledge that the fraction of afforestation was mentioned and discussed in previous studies, but none of these studies have explicitly demonstrated that it is a core factor that can reconcile the different approaches until our present work. In addition, the fraction change effect cannot cover the full range of the methodological differences. For example, we are unaware of any studies that have ever verified whether 'potential' effect could really be 'actualized'.

In response to the review comments, we will clarify the following points in the revised manuscript:

- (1) We acknowledge that the effect of fraction has been noticed and discussed in previous studies and that we are not the first study to examine this factor. We will emphasize the existing research on forest cover change effect on surface temperature change in the Introduction of the revised MS.
- (2) We will also explain that despite afforestation fraction was known to influence surface temperature change, whether it can fully explain the difference among different approaches has not been demonstrated.

(3) We will add a detailed description in the supplementary material to further explain the scope of methodological differences of different approaches, which include but are not limited to the fraction differences.

2. Second, the main finding is that the fraction of forestation (complete vs incomplete) explains the different magnitude of temperature effects. Fraction could indeed have a strong influence on the temperature signal. But it is not the only one. Other factors such as the timing of land cover change, length of the study period, and the spatial extent of forest cover change impact may also contribute.

(1) Taking the timing of de-/forestation as an example, if the change happened in the different years of the two periods of 2002–2004 (t1) and 2010–2014 (t2) (L277), changes in 2002 and 2010 would produce a larger temperature change compared to changes in 2004 and 2014, depending on whether the change signals lasted full three years or just the last year.

Sorry, but we are not completely sure what the reviewer refers to "timing of land cover change" and the example given is ambiguous. We appreciate and would like to provide a specific response if the reviewer gives us a more detailed description. If the reviewer is referring to "timing for a specific season in a given year", this is clearly not our focus because we are considering changes in the mean annual surface temperature.

As for "length of the study period", we admit that it can contribute to the magnitude of temperature effects, but the contribution was thought negligible here. For deforestation, e.g., logging and forest fire, conversion between forest and non-forest could be considered instant at the annual time scale of our research, as are the biophysical impacts induced by deforestation (Liu et al., 2018) (Fig. R1a, R1b). In contrast, afforestation often involves the succession of forests from a sparse canopy to a closed dense canopy until it can be observed by satellite as forest and very likely the change in surface temperature will follow the same pattern until it saturates in the closed-canopy forest (Fig. R1c). Global Forest Change dataset we used here defined forest gain as a stable closed canopy that can be distinguished from a nonforest state (Hansen et al., 2013), which gives us the confidence to conclude that the  $\Delta$ T signal has a good

chance of being saturated. In this case, the 'length of the study period' is expected to have little impact on our results.

We will briefly discuss and clarify these points in the Methods and Discussion sections in the revised MS to avoid similar confusion for future readers of the paper.

Figure R1. A conceptual scheme diagram showing the land surface temperature change following deforestation or afforestation. (a) Changes in annual mean land surface temperature (LST) following deforestation/afforestation. (b) The deforestation process could be considered as instant consisting of two clearly different stages. (c) Afforestation often leads to gradual forest succession or growth until it can be classified as stable forest cover by satellite data. Here it is represented with a three-stage process.

(2) More importantly, the space-for-time assumption is acceptable, but it is not strictly true in reality. The adjacent two sites did not share the same climate condition (see Chen 2016). This also contributes to the different temperature effects.

We admit that space-for-time is an assumption that cannot be verified on its own, which will inevitably result in uncertainties in the estimated  $\Delta T$ . But the consistency between 'potential' and 'actual' effects in our study proves that this assumption is broadly acceptable. These two points will be briefly discussed in the Discussion section in the revised MS.

(3) When the spatial extent of forest change is large, the local and nonlocal temperature effect

appear with heterogeneity which confounds the estimation of the local temperature.

In this study, the temperature effects based on the 'space-for-time', 'space-and-time' and SVD approaches strictly referred to the local effect, without considering any nonlocal effect (Duveiller et al., 2020, 2018; Winckler et al., 2019a). In fact, nonlocal effect is defined as biophysical effects due to changes in wide-ranging atmospheric circulation and advection of heat and moisture, which are triggered by afforestation (Duveiller et al., 2020; Fig. 2 in Pongratz et al., 2021; Fig. 1 in Winckler et al., 2019b). Within a searching window (e.g., 11km×11km in this study), any nonlocal effects cancel out when comparing temperature differences over these neighboring areas since advection and atmospheric circulation have similar effects on adjacent areas (Pongratz et al., 2021; Winckler et al., 2019a). Therefore, the effects derived in this study excluded nonlocal effects.

We will emphasize that biophysical effects here were "local effects" in the Method section in MS to avoid similar confusion for future readers of the paper.

(4) The consistency between the actual and potential effect is also scale-dependent. At small scales (e.g., 10m resolution), it would be easier to achieve full change compared to large scales (1km).

We agree that the realization of the full potential effect is scale-dependent and is more feasible at small scales. This comment is related to specific comments#5 and #7 below, to which we have responded with modified sentences. We will modify these statements in the revised MS.

Third, I feel the language of this manuscript should be improved and polished.

We will improve the language of this manuscript.

Specific comments:

1. L102-103 They may not assume 100% complete ground coverage. They used the defined forest and nonforest in the paper. Of course, due to inherent scaling and the mixed pixel issue

in remote sensing, the defined pixels cannot be 100% pure at a given scale. I think many studies were aware of this issue but they did not explicitly address it.

We revisited the related description in Duveiller et al. (2018) (Page 9): "The expected change in variable y associated with a transition from one vegetation type to another at the central pixel of the local window is then the difference between the  $y_p$  predicted for each pure vegetation type." Therefore, theoretically speaking the SVD approach quantified the 'full potential effect' by assuming transitions between land-cover types with 100% complete ground coverage, although pure vegetation type observed from satellite is hard to achieve. We prefer not to modify this statement in the MS.

**2. L161-162: How are the afforestation and adjacent control pixels defined?**

We will add a sentence in the Methods section in the revised MS to more clearly define afforestation and adjacent control pixels: "Here, pixels with  $F_{aff} > 0\%$  were defined as afforestation target pixels. A searching window of 11 km×11 km was then built centering on the afforestation pixel. Pixels with  $F_{aff}$  =0% within this searching window were defined as control pixels and were used to derive  $\Delta T_{res}$ ."

**3. L518: What do you mean "extensive variable"?**

We double-checked the concept of "extensive variable" in the existing literature (Scheider and Huisjes, 2019) and determined that this term should not be used here. Therefore, this word will be avoided from the revised MS. We will revise the related sentence in this section (L518) into: "This finding is in line with the fundamental fact that surface temperature at a given scale, can be strongly determined by the area fractions of its different components, with each component having a unique surface temperature, which also served as the theoretical foundation for the SVD technique to derive the full potential effect (Duveiller et al., 2018)."

4. L549 to 551: For this fractional dependency, it has been reported in such as Li 2016

We will cite the results from Li et al. (2016) to further support our research: "This is also consistent with a previous study which documented that these effects depend on the forest cover change thresholds used to define afforestation: the higher the threshold, the stronger is the impact on temperature (Li et al., 2016)."

5. L572-573: The actual and potential effect is also scale-dependent, and so is the feasibility of full afforestation in reality. Fully afforested could be easily achieved for a small pixel 30m. And for this pixel, the potential and actual could be similar following the findings of this work. At larger scales, it is more difficult to become "fully" afforested, which leads to larger differences between potential and actual impacts. Therefore, whether "achieving the full cooling potential" is scale-dependent.

We will revise this section (L572-573) into: "Full afforestation is often possible at small spatial scales, but at large scale it becomes challenging. So the realization of full potential effect by afforestation is scale-dependent."

6. L581-583: I disagree with the authors on this. The potential effect is useful as it measures the possible outcome of full conversion or mostly afforested (depending on resolution and scale), and whether it is realized depends on the fraction of the change. One can take into account the fractional change to convert the potential effect to more reasonable estimation. At least for this reason, it is not misleading. It is about different interpretation and clarification is needed.

We agree. We will revise this sentence into: "Potential cooling effects have a value in that they can serve to establish the envelope of effects and measure the possible outcome given the condition of full afforestation. However, given the challenge of full afforestation at large spatial scales, potential effects should be converted into a more reasonable estimate (i.e., actual effects) by taking into account the intensity of afforestation, to better represent possible policy ambitions and for the purpose of policy evaluation."

7. L602 to 605: I don't agree this statement because both the actual and potential effects are scale dependent. Without mentioning the scale, it is incorrect.

To avoid misunderstanding, we will revise these sentences into: "However, the realization of full potential effect is also scale-dependent. At small scales, full afforestation is more likely to occur, and consequently, potential impacts are more likely to be achieved, while full afforestation at large scale may not always be achievable, making it challenging to reach full potential impacts."

**References used in the responses:**

Alkama, R. and Cescatti, A.: Biophysical climate impacts of recent changes in global forest cover, Science, 351, 600–604, https://doi.org/10.1126/science.aac8083, 2016.

Chen, L. and Dirmeyer, P. A.: Adapting observationally based metrics of biogeophysical feedbacks from land cover/land use change to climate modeling, Environ. Res. Lett., 11, 034002, https://doi.org/10.1088/1748-9326/11/3/034002, 2016.

Duveiller, G., Caporaso, L., Abad-Viñas, R., Perugini, L., Grassi, G., Arneth, A., and Cescatti, A.: Local biophysical effects of land use and land cover change: towards an assessment tool for policy makers, Land Use Policy, 91, 104382, https://doi.org/10.1016/j.landusepol.2019.104382, 2020.

Duveiller, G., Hooker, J., and Cescatti, A.: The mark of vegetation change on Earth's surface energy balance, Nat Commun, 9, 679, https://doi.org/10.1038/s41467-017-02810-8, 2018.

Li, Y., Zhao, M., Mildrexler, D. J., Motesharrei, S., Mu, Q., Kalnay, E., Zhao, F., Li, S., and Wang, K.: Potential and Actual impacts of deforestation and afforestation on land surface temperature: IMPACTS OF FOREST CHANGE ON TEMPERATURE, J. Geophys. Res. Atmos., 121, 14,372-14,386, https://doi.org/10.1002/2016JD024969, 2016.

Liu, Z., Ballantyne, A. P., and Cooper, L. A.: Increases in Land Surface Temperature in Response to Fire in Siberian Boreal Forests and Their Attribution to Biophysical Processes, Geophys. Res. Lett., 45, 6485–6494, https://doi.org/10.1029/2018GL078283, 2018.

Scheider, S. and Huisjes, M. D.: Distinguishing extensive and intensive properties for meaningful geocomputation and mapping, International Journal of Geographical Information Science, 33, 28–54, https://doi.org/10.1080/13658816.2018.1514120, 2019.

Hansen, M. C., Potapov, P. V., Moore, R., Hancher, M., Turubanova, S. A., Tyukavina, A., Thau,D., Stehman, S. V., Goetz, S. J., and Loveland, T. R.: High-resolution global maps of 21stcentury forest cover change, science, 342, 850–853, 2013.

Pongratz, J., Schwingshackl, C., Bultan, S., Obermeier, W., Havermann, F., and Guo, S.: Land Use Effects on Climate: Current State, Recent Progress, and Emerging Topics, Curr Clim Change Rep, 7, 99–120, https://doi.org/10.1007/s40641-021-00178-y, 2021.

Winckler, J., Lejeune, Q., Reick, C. H., and Pongratz, J.: Nonlocal Effects Dominate the Global Mean Surface Temperature Response to the Biogeophysical Effects of Deforestation, Geophys. Res. Lett., 46, 745–755, https://doi.org/10.1029/2018GL080211, 2019a.

Winckler, J., Reick, C. H., Bright, R. M., and Pongratz, J.: Importance of Surface Roughness for the Local Biogeophysical Effects of Deforestation, J. Geophys. Res. Atmos., 124, 8605–8618, https://doi.org/10.1029/2018JD030127, 2019b.

---

## Author Comment (AC3)

Ref.: MS. bg-2022-317 Biogeosciences Reconciling different approaches to quantifying land surface temperature impacts of afforestation using satellite observations

**Reviewer#2**

General consideration:

The biophysical effects of deforestation/afforestation have drawn a lot of attention in the past few years. However, the results are not very consistent among different studies using different products and methods. The authors revealed the methodological differences among different studies and summarized them into one actual and two potential temperature effects. They also used afforestation in China as a test case to quantify the differences in biophysical effects using the three approaches and verify their hypotheses. The manuscript is well-structured, and the results are clearly represented. I would recommend the publication of this manuscript after minor revisions.

Some minor comments: Language needs to be further polished throughout the text. Some long sentences are difficult to understand.

We thank Reviewer#2 for the positive comments which allow us to improve our manuscript. Please find below our detailed responses to the review comments, with original comments in black and our responses in blue.

**Specific comments:**

1. L30, "and that it ... explained", Not clear.

To avoid any potential confusion, we will modify the sentence as "The magnitude of  $\Delta T_a$  increases with the fraction of the pixel actually afforested (Faff) and Faff explained 89% of the variation in  $\Delta T_a$ ."

2. In Methods, need to clarify how gridded effects were aggregated into the country mean for

comparison among the three approaches, because different LC/LST data may have different coverage. How is the overlapped region representative for the whole country?

We verified the representativeness of our research samples by examining the distributions of the temperature effects from original pixels and research pixels (i.e., spatial overlapped samples of different approaches). Fig. R1 shows that 17.5% of original samples for actual effects were retained for further analysis and preserved the same mean value (-0.07 K); while 20.2% of original samples for potential effects, with the mean value (-0.64 K) being close to the mean value of all samples (-0.42 K). The results of the original samples are similar to that of the research samples in the Manuscript (MS) (Fig. 5 and Fig. 7 in MS). Therefore, we believe that it is acceptable to use these overlapped samples as research samples in this study. Although we have verified these research samples' representativeness of the whole country, we still need to briefly claim that being representative is not our research objective; instead, we need these samples to compare different approaches.

To address the first half of this comment, we will add related clarifications in the Method section in the revised MS: "First, we limited the analysis to only those pixels shared by all three approaches, and this resulted in 96058 sample pixels at 1km resolution. These spatially overlapping samples maintained the distributional characteristics of the original samples with similar outcomes (Fig. R1). Then, the average values of three approaches were calculated and compared."

Figure R1. (a) Histogram of  $\Delta T_a$  of all pixels based on GFC dataset (b) Histogram of  $\Delta T_a$  for research samples. (c) Histogram of  $\Delta T_{p1}$  of all pixels based on GFC dataset (d) Histogram of  $\Delta T_{p2}$  for research samples.

3. L275-277, afforestation from GFC is not consistent with the inventory data, so can the results based on GFC be considered as the real biophysical effects of afforestation in China? I think this key message is important for policy makers.

The central objective of our study is to demonstrate the fraction of afforestation is a core factor that can reconcile different approaches. Thus, this question is a little out of our scope, but we addressed it in greater detail below.

We believe this question is related to the accuracy of afforestation from Global Forest Change (GFC). According to Hansen et al. (2013), considerable forest growth in China was not easily detected in time-series of satellite imagery (i.e., GFC) when compared to forest inventory assessed in FAO Forest Resource Assessment (FRA). This discrepancy may arise from the definition of 'forest', classification system, spatial resolution, and algorithm (Chen et al., 2020). Nevertheless, the GFC product shows an overall accuracy greater than 99% at the global scale for the observed forest gained area when it was compared with forest area statistics reported in FRA, LiDAR detection (Geoscience Laser Altimetry System), and MODIS NDVI time series. Therefore, GFC was recommended to be utilized in forest and forest change estimates (Chen et al., 2020).

**al., 2020).**

In this study, the net forest gain area is about 24,372 km2 based on GFC, while the overlapped region included in this research is about 1,400 km2 (Fig. R2), both of which are significantly lower than 157,000 km2 as indicated by National Forest Resources Inventory (SFA, 2014). We thus cannot give a precise evaluation of the actual biophysical effects of afforestation in China. Nevertheless, based on the analysis (Fig. R1), the distribution of research samples was similar to the original distribution on each afforestation intensity bin and maintained the same overall actual effect of -0.07 K.

To address this comment, we will provide a description of GFC's accuracy in the Method section and briefly claim that "this study cannot provide a precise actual effect of afforestation in China" in the Discussion section of the revised MS.

Figure R2. (a) Histogram of the afforestation intensity (%) based on net forest gain from GFC dataset (b) Histogram of afforestation intensity (%) from research samples.

4. L391, that's what I meant, the afforestation area is much smaller than the national inventory.

The specific reasons can be found in our responses to comment #3. Although this question is out of our research scope, we still clarified these two points on this question: Firstly, GFC was appropriate for use in detecting afforestation (Chen et al., 2020). Secondly, despite the result based on GFC cannot provide an accurate assessment of the actual biophysical effects of

afforestation in China, this does not impede our understanding of the actual effects of afforestation (Fig. R1a, b). We will briefly discuss these in the Discussion section of the revised MS.

5. Fig. 4, better to show the latitudes on the left axis of (a)

We will add legends to Fig. 6.

6. Fig. 5, did you consider the spatial distribution of each bin? Whether the regions with higher  $F_{aff}$  happen to be in the tropics with larger cooling effects?

To address the reviewer's comment, we checked the  $\Delta T_a$  within each afforestation bin in different climate zones (Fig. R3). On average, afforestation in the tropical zone had the strongest cooling effect, followed by the subtropics zone, temperate zone, and Qinghai-Tibet Plateau. Such climate zone patterns on the effect induced by afforestation have been reported by previous studies that forest restoration contributes to the surface cooling in tropical zones whilst minor warming might occur in boreal forest zones (Alkama and Cescatti, 2016; Li et al., 2015; Peng et al., 2014). More specifically, the cooling effect was stronger in the tropical zone than in other zones with the same afforestation intensity, which is consistent with our expectation since the enhanced evapotranspiration in the tropical would release more latent heat when afforestation with fixed intensity occurred than other regions with the same intensity (Li et al., 2016).

We will add discussion on "cooling effect in different climate zones with the same afforestation intensity" in the Discussion section and add Figure R3 in Supplemental Material.

Figure R3.  $\Delta T_a$  within each afforestation intensity (Faff) bin over four climate zones (Tropics, Subtropics, Temperate and Qinghai-Tibet Plateau Plateau) in China. The climate zone was based on Climate Regionalization of China (https://www.resdc.cn/data.aspx?DATAID=243).

7. Fig. 8, I guess the differences for changes in seasonal fluxes would be much larger between the partial and full coverage of each pixel, especially in the snowing regions in winter.

We agree. Previous research has documented that in high-latitude regions the snow-covered short vegetation has larger albedo than forest in spring and winter, leading to a greater warming effect in the transition from openland to forest (Peng et al., 2014; Li et al., 2015; Lawrence et al., 2022). In our study, it is expected that the difference in seasonal fluxes between mixed potential (i.e., effects of partial coverage of pixels) and full potential effect was much greater, given that full transition can significantly amplify the albedo-induced warming effect at high latitude. Here, in the specific instance of shortwave radiation (SWout), we added some seasonal flux analysis for the summer (June to August) and winter (December to February) seasons, respectively (Fig. R4 and R5). Fig. R5 shows that the magnitude of the full potential SWout effect was stronger than the mixed potential effect (Fig. R4). In winter, there was a strong decrease in SWout than in summer, and the decrease was larger for the boreal forest areas northward 45°N than the lower latitudinal areas southward 45°N due to the snow cover in the forest understory (Fig. R5).

We believe this part will supplement our results; therefore, this point will be briefly discussed

in the Discussion section of the revised manuscript and Fig. R5 will be added in the Supplemental Material.

---

## Author Response (AR1)

**Response to the Editor and Reviewers**

Dear Editor and reviewers,

Thank you very much for giving us this valuable opportunity to revise the manuscript and thank two anonymous reviewers' comments and Dr. Chao Zhang's comments, which helped us improve our manuscript.

The language of the revised manuscript has been improved by Dr. Alistair David Culf, an expert in Scientific Writing & Editing. According to the comments, we have revised the manuscript carefully and highlighted the revised portions in blue. The point-by-point responses to the editor and reviewer comments are listed as follows (in blue).

Thank you for your kind consideration. We are looking forward to your favorable decision.

Yours Faithfully and on behalf of all authors,

Huanhuan Wang

**Editor's comments:**

Thank you for taking the time to draft comprehensive responses to the two reviewers and the detailed comments from Dr. Chao Zhang. I agree with the reviewers that this paper highlights an important factor affecting the result of afforestation temperature effect studies, and could be a welcome addition to the literature after suitable revision. However, I also agree with the spirit of Dr. Chao Zhang's comment 6 and reviewer 1's comment 2 – there are a number of other factors at work here, and these should also be clearly explained in the paper for the sake of clarity/to avoid misleadingly making it seem like the afforestation fraction is the only relevant factor. Indeed, even in Figure 7, there are still significant differences between the three right-most bar. In the responses to these comments, you propose to add a discussion of these issues to the supplementary material, but I suggest you try to fit it into the main manuscript for clarity. Please proceed to upload a revised manuscript.

Thanks for the helpful advice. We have uploaded the manuscript according to the suggestions. The objective of this manuscript has been clarified as "highlighting the role of afforestation fraction in reconciling afforestation effects by different methods". Regarding Dr. Chao Zhang's comment 6 and reviewer 1's comment 2, we fully agree that there are other factors contributing to the differences in temperature effects produced by different methods. As suggested by the editor, we have added additional reviewer-suggested factors to the Discussion section of the manuscript to clearly clarify that afforestation fraction cannot cover the full range of the temperature effects differences.

**Reviewer#1**

**General consideration:**

The manuscript "Reconciling different approaches to quantifying land surface temperature impacts of afforestation using satellite observations" by Wang et al presented thoughtful analyses regarding three different types of temperature effects of forestation that appeared in the literature (potential vs actual) and trying to explain the causes of the different magnitudes. The research is a nice addition to the literature on this topic as it is helpful to clarify the interpretation of different results.

We thank Reviewer#1 for the general positive comments and confirming the additional value of our work to the existing literature.

**Major comments:**

1. First, I disagree with the authors' interpretation of these results and the claim that the causes of the different estimates are unknown. On the contrary, spatial scale or fractions of forest change matters for interpreting the temperature impact, which has been considered in previous studies. Taking the influential work cited by the authors as an example:

In Alkama 2016, the fraction of forest cover change is explicitly taken into account, and the results clearly indicated that the temperature effect depended on the fraction of change.

In Li 2016, the fractional dependency has been reported: "It should be noted that the estimated impacts also depend on the thresholds used to define forest cover change, as discussed in section 2.2. The sensitivity analysis shows that a higher threshold to define forest change leads to stronger impacts on temperature."

In Duveiller 2018, they used the temperature effect of 100% conversion to avoid the influence of fractional changes.

The strength of this work is that it explicitly addressed this question. Perhaps the authors could consider an alternative title better reflecting this point.

Maybe some improper expressions (very likely they could be lines 25 and 81) in our original manuscript (MS) made the reviewer conclude that we have claimed that "the causes of the different estimates are unknown." But in fact, we did not explicitly claim this. Instead, we suspected that different forest cover changes in different approaches could potentially influence the LST change, which is one of the motivations for this study.

The reviewer seems to imply that the fraction of forest cover change was known as the cause of different magnitudes of LST change in previous studies. But this is not true. We acknowledge that the fraction of afforestation was mentioned and discussed in previous studies, but none of these studies have explicitly demonstrated that it is a core factor that can reconcile the different approaches until our present work. In addition, the fraction change effect cannot cover the full range of the methodological differences. For example, we are unaware of any studies that have ever verified whether 'potential' effect could really be 'actualized'.

In response to the review comments, we clarified the following points in the revised manuscript (MS):

(1) We emphasized the existing research on forest cover change effect on surface temperature change in the revised text lines 101-104: "Previous studies have revealed the fraction of forest change as an important factor determining the magnitude of the afforestation effect. Alkama and Cescatti (2016) indicated that the actual temperature effect is fraction-dependent, and Li et al. (2016a) pointed out that use of a higher threshold to define forest change resulted in a stronger potential effect."

- (2) Despite afforestation fraction was known to influence surface temperature change, whether it can fully explain the difference among different approaches has not been demonstrated. We have revised related statements, and details can be found in Lines 22-25, Lines 77-78, and Lines 105-109.
- (3) We further explained the scope of methodological differences of different approaches, which include but are not limited to afforestation fraction differences. This point is detailed in Lines 573-583.

2. Second, the main finding is that the fraction of forestation (complete vs incomplete) explains the different magnitude of temperature effects. Fraction could indeed have a strong influence on the temperature signal. But it is not the only one. Other factors such as the timing of land cover change, length of the study period, and the spatial extent of forest cover change impact may also contribute.

(1) Taking the timing of de-/forestation as an example, if the change happened in the different years of the two periods of 2002–2004 (t1) and 2010–2014 (t2) (L277), changes in 2002 and 2010 would produce a larger temperature change compared to changes in 2004 and 2014, depending on whether the change signals lasted full three years or just the last year.

Sorry, but we are not completely sure what the reviewer refers to "timing of land cover change" and the example given is ambiguous. We appreciate and would like to provide a specific response if the reviewer gives us a more detailed description. If the reviewer is referring to "timing for a specific season in a given year", this is clearly not our focus because we are considering changes in the mean annual surface temperature.

As for "length of the study period", we admit that it can contribute to the magnitude of temperature effects. For deforestation, e.g., logging and forest fire, conversion between

forest and non-forest could be considered instant at the annual time scale of our research, as are the biophysical impacts induced by deforestation (Liu et al., 2018) (Fig. R1a, R1b). In contrast, afforestation often involves the succession of forests from a sparse canopy to a closed dense canopy until it can be observed by satellite as forest and very likely the change in surface temperature will follow the same pattern until it saturates in the closed-canopy forest (Fig. R1c). Global Forest Change dataset we used here defined forest gain as a stable closed canopy that can be distinguished from a nonforest state (Hansen et al., 2013), which gives us the confidence to conclude that the  $\Delta T$  signal has a good chance of being saturated. But differences in surface temperatures may still exist between newly established forests and the mature existing forests that were used in the 'potential effect' approaches. Thus, we cannot exclude the possible contribution of such a mechanism to the difference between the actual and potential effects, which failed to be reconciled. We have summarized these points in the Discussion sections (Lines 585-597) in the revised MS to avoid similar confusion for future readers of the paper.

Figure R1. A conceptual scheme diagram showing the land surface temperature change following deforestation or afforestation. (a) Changes in annual mean land surface temperature (LST) following deforestation/afforestation. (b) The deforestation process could be considered as instant consisting of two clearly different stages. (c) Afforestation

often leads to gradual forest succession or growth until it can be classified as stable forest cover by satellite data. Here it is represented with a three-stage process.

(2) More importantly, the space-for-time assumption is acceptable, but it is not strictly true in reality. The adjacent two sites did not share the same climate condition (see Chen 2016). This also contributes to the different temperature effects.

We admit that space-for-time is an assumption that cannot be verified on its own, which will inevitably result in uncertainties in the estimated  $\Delta T$ . But the consistency between 'potential' and 'actual' effects in our study proves that this assumption is broadly acceptable. These two points have been added in the revised text lines of 580-583.

(3) When the spatial extent of forest change is large, the local and nonlocal temperature effect appear with heterogeneity which confounds the estimation of the local temperature.

In this study, the temperature effects based on the 'space-for-time', 'space-and-time' and SVD approaches strictly referred to the local effect, without considering any nonlocal effect (Duveiller et al., 2020, 2018; Winckler et al., 2019a). In fact, nonlocal effect is defined as biophysical effects due to changes in wide-ranging atmospheric circulation and advection of heat and moisture, which are triggered by afforestation (Duveiller et al., 2020; Fig. 2 in Pongratz et al., 2021; Fig. 1 in Winckler et al., 2019b). Within a searching window (e.g., 11km×11km in this study), any nonlocal effects cancel out when comparing temperature differences over these neighboring areas since advection and atmospheric circulation have similar effects on adjacent areas (Pongratz et al., 2021; Winckler et al., 2019a). Therefore, the effects derived in this study excluded nonlocal effects. This point has also been summarized in the revised text lines of 258-262 to avoid any potential confusion.

(4) The consistency between the actual and potential effect is also scale-dependent. At small scales (e.g., 10m resolution), it would be easier to achieve full change compared to large scales (1km).

We agree that the realization of the full potential effect is scale-dependent and is more feasible at small scales. We have modified related statements in the revised MS, as follows:

We have revised the section (Lines 572-573), which now reads (Lines 616-618): "Full afforestation is often possible at small spatial scales but becomes challenging at large scale. Therefore, the realization of the full potential effect by afforestation is scale-dependent."

The section comprising Lines 602-605 now reads as (Lines 649-653): "Potential cooling effects have a value in academic studies where they can be used to establish an envelope of effects, but their realization at large scales is challenging given the scale dependency. The reconciliation of the different approaches demonstrated here stresses that the afforestation fraction should be accounted for in order to bridge different estimates of surface cooling effects in policy evaluation."

Third, I feel the language of this manuscript should be improved and polished.

The revised manuscript has been edited and improved by Dr. Alistair David Culf, an expert in Scientific Writing & Editing.

**Specific comments:**

1. L102-103 They may not assume 100% complete ground coverage. They used the defined forest and nonforest in the paper. Of course, due to inherent scaling and the mixed pixel issue in remote sensing, the defined pixels cannot be 100% pure at a given scale. I

think many studies were aware of this issue but they did not explicitly address it.

We revisited the related description in Duveiller et al. (2018) (Page 9): "The expected change in variable y associated with a transition from one vegetation type to another at the central pixel of the local window is then the difference between the  $y_p$  predicted for each pure vegetation type." Therefore, theoretically speaking the SVD approach quantified the 'full potential effect' by assuming transitions between land-cover types with 100% complete ground coverage, although pure vegetation type observed from satellite is hard to achieve. We retained this statement in the MS.

**2. L161-162: How are the afforestation and adjacent control pixels defined?**

We added a sentence in the Methods section (Lines 163-166) in the revised MS to more clearly define afforestation and adjacent control pixels: "Here, pixels with  $F_{aff} > 0\%$  were defined as afforestation target pixels. A searching window of 11 km by 11 km was established, centered on the afforestation pixel. Within this window, pixels with  $F_{aff} = 0\%$  were defined as control pixels and were used to derive  $\Delta T_{res}$ ."

**3. L518: What do you mean "extensive variable"?**

We double-checked the concept of "extensive variable" in the existing literature (Scheider and Huisjes, 2019) and realized that this term cannot be used here. Therefore, this word has been removed in the revised MS. We have revised the related sentences and the details can be found in Lines 534-537: "This result is consistent with the fact that the observed temperature for a mixed surface is determined by the area fractions of its respective components, with each component having a unique temperature. This fact also forms the theoretical foundation for the SVD technique used to derive the full potential effect (Duveiller et al., 2018)."

**4. L549 to 551: For this fractional dependency, it has been reported in such as Li 2016**

We have cited the results from Li et al. (2016) to further support our research and the details can be found in Lines 566-569: "This is consistent with the finding of a previous study on the dependence of the temperature effect on the forest cover change thresholds that were used to define afforestation: the higher the threshold, the stronger the impact on temperature (Li et al., 2016)."

5. L572-573: The actual and potential effect is also scale-dependent, and so is the feasibility of full afforestation in reality. Fully afforested could be easily achieved for a small pixel 30m. And for this pixel, the potential and actual could be similar following the findings of this work. At larger scales, it is more difficult to become "fully" afforested, which leads to larger differences between potential and actual impacts. Therefore, whether "achieving the full cooling potential" is scale-dependent.

We have revised the section (Lines 572-573), which now reads (Lines 617-619): "Full afforestation is often possible at small spatial scales but becomes challenging at large scale. Therefore, the realization of the full potential effect by afforestation is scale-dependent."

6. L581-583: I disagree with the authors on this. The potential effect is useful as it measures the possible outcome of full conversion or mostly afforested (depending on resolution and scale), and whether it is realized depends on the fraction of the change. One can take into account the fractional change to convert the potential effect to more reasonable estimation. At least for this reason, it is not misleading. It is about different interpretation and clarification is needed.

We agree. We revised this sentence (Lines 581-583), which now reads as (Lines 626-630): "Potential cooling effects have a value in that they can serve to establish the envelope of effects and measure possible outcomes given the condition of full afforestation. However, given the challenge of full afforestation at large spatial scales, potential effects should be converted into a more realistic estimate (i.e., actual effects), by taking into account the intensity of afforestation, to better represent policy ambitions."

**7. L602 to 605: I don't agree this statement because both the actual and potential effects are scale dependent. Without mentioning the scale, it is incorrect.**

To avoid misunderstanding, we revised these sentences (Lines 602-605), which now reads as (lines 649-653): "Potential cooling effects have a value in academic studies where they can be used to establish an envelope of effects, but their realization at large scales is challenging given its nature of scale dependency. The reconciliation of different approaches demonstrated here stresses that the afforestation fraction should be accounted for in bridging different estimations of surface cooling effects in policy evaluation."

**References used in the responses:**

Alkama, R. and Cescatti, A.: Biophysical climate impacts of recent changes in global forest cover, Science, 351, 600–604, https://doi.org/10.1126/science.aac8083, 2016.

Chen, L. and Dirmeyer, P. A.: Adapting observationally based metrics of biogeophysical feedbacks from land cover/land use change to climate modeling, Environ. Res. Lett., 11, 034002, https://doi.org/10.1088/1748-9326/11/3/034002, 2016.

Duveiller, G., Caporaso, L., Abad-Viñas, R., Perugini, L., Grassi, G., Arneth, A., and Cescatti, A.: Local biophysical effects of land use and land cover change: towards an assessment tool for policy makers, Land Use Policy, 91, 104382,

https://doi.org/10.1016/j.landusepol.2019.104382, 2020.

Duveiller, G., Hooker, J., and Cescatti, A.: The mark of vegetation change on Earth's surface energy balance, Nat Commun, 9, 679, https://doi.org/10.1038/s41467-017-02810-8, 2018.

Li, Y., Zhao, M., Mildrexler, D. J., Motesharrei, S., Mu, Q., Kalnay, E., Zhao, F., Li, S., and Wang, K.: Potential and Actual impacts of deforestation and afforestation on land surface temperature: IMPACTS OF FOREST CHANGE ON TEMPERATURE, J. Geophys. Res. Atmos., 121, 14,372-14,386, https://doi.org/10.1002/2016JD024969, 2016.

Liu, Z., Ballantyne, A. P., and Cooper, L. A.: Increases in Land Surface Temperature in Response to Fire in Siberian Boreal Forests and Their Attribution to Biophysical Processes, Geophys. Res. Lett., 45, 6485–6494, https://doi.org/10.1029/2018GL078283, 2018.

Scheider, S. and Huisjes, M. D.: Distinguishing extensive and intensive properties for meaningful geocomputation and mapping, International Journal of Geographical Information Science, 33, 28–54, https://doi.org/10.1080/13658816.2018.1514120, 2019.

Hansen, M. C., Potapov, P. V., Moore, R., Hancher, M., Turubanova, S. A., Tyukavina, A., Thau, D., Stehman, S. V., Goetz, S. J., and Loveland, T. R.: High-resolution global maps of 21st-century forest cover change, science, 342, 850–853, 2013.

Pongratz, J., Schwingshackl, C., Bultan, S., Obermeier, W., Havermann, F., and Guo, S.: Land Use Effects on Climate: Current State, Recent Progress, and Emerging Topics, Curr Clim Change Rep, 7, 99–120, https://doi.org/10.1007/s40641-021-00178-y, 2021.

Winckler, J., Lejeune, Q., Reick, C. H., and Pongratz, J.: Nonlocal Effects Dominate the

Global Mean Surface Temperature Response to the Biogeophysical Effects of Deforestation, Geophys. Res. Lett., 46, 745–755, https://doi.org/10.1029/2018GL080211, 2019a.

Winckler, J., Reick, C. H., Bright, R. M., and Pongratz, J.: Importance of Surface Roughness for the Local Biogeophysical Effects of Deforestation, J. Geophys. Res. Atmos., 124, 8605–8618, https://doi.org/10.1029/2018JD030127, 2019b.

**Reviewer#2**

**General consideration:**

The biophysical effects of deforestation/afforestation have drawn a lot of attention in the past few years. However, the results are not very consistent among different studies using different products and methods. The authors revealed the methodological differences among different studies and summarized them into one actual and two potential temperature effects. They also used afforestation in China as a test case to quantify the differences in biophysical effects using the three approaches and verify their hypotheses. The manuscript is well-structured, and the results are clearly represented. I would recommend the publication of this manuscript after minor revisions.

Language needs to be further polished throughout the text. Some long sentences are difficult to understand.

We thank Reviewer#2 for the positive comments which allow us to improve our manuscript. We have simplified the expressions in the revised manuscript when possible. The revised manuscript has been edited and improved by Dr. Alistair David Culf, a native English-speaking scientist.

**Specific comments:**

1. L30, "and that it ... explained", Not clear.

To avoid any potential confusion, we have modified the sentence as " $\Delta T_a$  increased with the afforestation fraction, which explained 89% of its variation."

2. In Methods, need to clarify how gridded effects were aggregated into the country mean for comparison among the three approaches, because different LC/LST data may have different coverage. How is the overlapped region representative for the whole country?

We verified the representativeness of our research samples by examining the distributions of the temperature effects from original pixels and research pixels (i.e., spatial overlapped samples of different approaches). Fig. R1 shows that 17.5% of original samples for actual effects were retained for further analysis and preserved the same mean value (-0.07 K); while 20.2% of original samples for potential effects, with the mean value (-0.64 K) being close to the mean value of all samples (-0.42 K). The results of the original samples are similar to that of the research samples in the Manuscript (MS) (Fig. 5 and Fig. 7 in MS). Therefore, we believe that it is acceptable to use these overlapped samples as research samples in this study. Although we have verified these research samples' representativeness of the whole country, we still need to briefly claim that being representative is not our research objective; instead, we need these samples to compare different approaches.

We have supplemented the Appendix with Figure R1 and added related clarifications into our revised manuscript and details can be found in Lines 379-383: "The spatial distributions of original samples for the three methods are different because of the different land-cover maps used (Fig. 2 and Figure A1) and, therefore, the statistical analysis was limited to those pixels shared by all three approaches: 96,058 sample pixels at 1km resolution. The distribution of these shared sample pixels retained the characteristics of the spatial distribution of the original samples (Figure A2)."

Figure R1. (a) Histogram of  $\Delta T_a$  of all pixels based on GFC dataset (b) Histogram of  $\Delta T_a$  for research samples. (c) Histogram of  $\Delta T_{p1}$  of all pixels based on GFC dataset (d) Histogram of  $\Delta T_{p2}$  for research samples.

3. L275-277, afforestation from GFC is not consistent with the inventory data, so can the results based on GFC be considered as the real biophysical effects of afforestation in China? I think this key message is important for policy makers.

We believe this question is related to the accuracy of afforestation from Global Forest Change (GFC). According to Hansen et al. (2013), considerable forest growth in China was not easily detected in time-series of satellite imagery (i.e., GFC) when compared to forest inventory assessed in FAO Forest Resource Assessment (FRA). This discrepancy may arise from the definition of 'forest', classification system, spatial resolution, and algorithm (Chen et al., 2020). Nevertheless, the GFC product shows an overall accuracy greater than 99% at the global scale for the observed forest gained area when it was compared with forest area statistics reported in FRA, LiDAR detection (Geoscience Laser Altimetry System), and MODIS NDVI time series. Therefore, GFC was recommended to be utilized in forest and forest change estimates (Chen et al., 2020).

In this study, the net forest gain area is about 24,372 km2 based on GFC, while the overlapped region included in this research is about 1,400 km2 (Fig. R2), both of which are significantly lower than 157,000 km2 as indicated by National Forest Resources Inventory (SFA, 2014). We thus cannot give a precise evaluation of the actual biophysical effects of afforestation in China. Nevertheless, based on the analysis (Fig. R1), the distribution of research samples was similar to the original distribution on each afforestation intensity bin and maintained the same overall actual effect of -0.07 K.

Although we addressed this comment in greater detail above, this question is a little out of our research scope, and the central objective of our study is to demonstrate that the fraction of afforestation is a core factor that can reconcile different approaches. The above description of GFC's accuracy is summarized in the revised text Lines 327-331 to prove that GFC is appropriate for detecting afforestation.

Figure R2. (a) Histogram of the afforestation intensity (%) based on net forest gain from

---

## Referee Report (RR1)

I would like to clarify my original comment:

 Taking the timing of de-/forestation as an example, if the change happened in the different years of the two periods of 2002–2004 (t1) and 2010–2014 (t2) (L277), changes in 2002 and 2010 would produce a larger temperature change compared to changes in 2004 and 2014, depending on whether the change signals lasted full three years or just the last year.

Here I provide an example to show the impact of the timing of the land cover change (figure attached).

Suppose LST for land type A is 10 and for land type B is 20. A land cover change happened in a year in the second period (2010-2014). When it happened in 2014 (the last year of period 2), the LST change between the mean LST of the two periods would be 2, and when it happened in 2010 (the first year of period 2), the LST change would be 10.

| A              | В    | С    | D    | E    | F    | G    | Н    | 1    | J              | K         | L          |
|----------------|------|------|------|------|------|------|------|------|----------------|-----------|------------|
|                | 2002 | 2003 | 2004 | 2010 | 2011 | 2012 | 2013 | 2014 | mean_2002_2004 | mean_2010 | LST_change |
| change in 2014 | 10   | 10   | 10   | 10   | 10   | 10   | 10   | 20   | 10             | 12        | 2          |
| change in 2010 | 10   | 10   | 10   | 20   | 20   | 20   | 20   | 20   | 10             | 20        | 10         |
|                |      |      |      |      |      |      |      |      |                |           |            |

2) The comparison of methods for potential and actual impacts is based on scenario of afforestation, I wonder if the conclusions can be generalized to deforestation or other the land cover change impact?

---

## Author Response (AR2)

Ref.: MS. bg-2022-317 Biogeosciences Reconciling different approaches to quantifying land surface temperature impacts of afforestation using satellite observations

**Reviewer#1**

I would like to clarify my original comment:

1) Taking the timing of de-/forestation as an example, if the change happened in the different years of the two periods of 2002–2004 (t1) and 2010–2014 (t2) (L277), changes in 2002 and 2010 would produce a larger temperature change compared to changes in 2004 and 2014, depending on whether the change signals lasted full three years or just the last year.

Here I provide an example to show the impact of the timing of the land cover change (figure attached). Suppose LST for land type A is 10 and for land type B is 20. A land cover change happened in a year in the second period (2010-2014). When it happened in 2014 (the last year of period 2), the LST change between the mean LST of the two periods would be 2, and when it happened in 2010 (the first year of period 2), the LST change would be 10.

Thanks for the clarification on this point by the reviewer. We provide further explanations on this issue and have made according changes in the revised manuscript.

We understand the case provided by the reviewer and agree that the timing of the land cover change can influence the quantified LST changes by our approach, i.e., by averaging the LST for the several years of our starting and end period and looking at their differences. But we also argue that the magnitude of such influences depends on the type of the land cover change concerned. Deforestation typically involves rapid land-use transitions, and the resulting temperature effects are almost instant (Liu et al., 2018). Here, the instantaneous LST change

at the annual time scale provided by the reviewer represents more likely a deforestation process. In this case we agree that our time-averaging approach will cause the error in the quantified  $\Delta$ LST as shown in the example provided by the reviewer.

In contrast, afforestation often involves the succession of forests from a sparse canopy to a closed canopy until the newly established forest can be reliably observed by satellite. Accordingly, the biophysical effect will follow the same pattern until it saturates in the closed-canopy forest (please refer to the Figure R1 in our previous responses to the reviewers' comments). Indeed, observation studies show that closed dense-canopy old forests can exert greater cooling effect than the open-canopy young forests (Zhang et al., 2021; Windisch et al., 2021). Hence, given the gradual nature of the afforestation effect on LST, when we quantify the afforestation effect by comparing the time-averaging LST before and after afforestation, the influence of the specific 'timing of afforestation' is expected to be small.

Following the explanation above, the effect of 'timing of land cover change' has been clarified in Lines 586–601 in our revised manuscript: "Differences between the actual and potential temperature effects can also arise from the influences of both the timing of the afforestation and the time length elapsed following afforestation. However, such influences are expected to be small in our study. We argue that such influences should be more pronounced in the case of deforestation than afforestation. The temperature effect caused by deforestation is considered to be instant (Liu et al., 2018). As a result, if deforestation occurred in one specific year of our starting time window (i.e., 2002–2004), using the time-averaging LST over the whole time window to represent the LST before deforestation will greatly bias the quantified  $\Delta T$ . In contrast, afforestation-driven surface temperature change can only gradually increase with forest development. The LST effect depends on different forest development stages and is expected to saturate only when the forest canopy stabilizes

(Zhang et al., 2021; Windisch et al., 2021). Observation studies show that closed densecanopy old forests can exert greater cooling effect than the open-canopy young forests (Zhang et al., 2021; Windisch et al., 2021). Hence, given the gradual nature of the afforestation effect on LST, when we quantify the afforestation effect by comparing the timeaveraging LST before and after afforestation, the influence of the specific 'timing of afforestation' is expected to be small."

2) The comparison of methods for potential and actual impacts is based on scenario of afforestation, I wonder if the conclusions can be generalized to deforestation or other the land cover change impact?

We believe in principle both our methods and conclusions should be applicable to deforestation and other land cover change impacts. However, larger uncertainty may arise as we explained in our response to Comment #1 by the reviewer, that the "timing of the land cover change" plays a role in the quantified biophysical impacts by approach, especially for the scenario in which instant temperature effects by land cover conversions expected, such as deforestation. Hence, we limited our conclusion to the scenario of afforestation, which is also within the scope of this study.

**References**

Liu, Z., Ballantyne, A. P., and Cooper, L. A.: Increases in Land Surface Temperature in Response to Fire in Siberian Boreal Forests and Their Attribution to Biophysical Processes, Geophys. Res. Lett., 45, 6485–6494, https://doi.org/10.1029/2018GL078283, 2018.

Windisch, M. G., Davin, E. L., and Seneviratne, S. I.: Prioritizing forestation based on biogeochemical and local biogeophysical impacts, Nat. Clim. Chang., 11, 867–871, https://doi.org/10.1038/s41558-021-01161-z, 2021.

Zhang, Z., Zhang, F., Wang, L., Lin, A., and Zhao, L.: Biophysical climate impact of forests with different age classes in mid- and high-latitude North America, Forest Ecology and Management, 494, 119327, https://doi.org/10.1016/j.foreco.2021.119327, 2021.